# WRN promotes bone development and growth by unwinding SHOX-G-quadruplexes via its helicase activity in Werner Syndrome

Yuyao Tian[1], Wuming Wang[1,2], Sofie Lautrup [3], Hui Zhao [1,4], Xiang Li[5], Patrick Wai Nok Law[1], Ngoc-Duy Dinh [6], Evandro Fei Fang[3], Hoi Hung Cheung[1] & Wai-Yee Chan [1,2,4] ✉

Werner Syndrome (WS) is an autosomal recessive disorder characterized by premature aging due to mutations of the *WRN* gene. A classical sign in WS patients is short stature, but the underlying mechanisms are not well understood. Here we report that *WRN* is indispensable for chondrogenesis, which is the engine driving the elongation of bones and determines height. Zebrafish lacking *wrn* exhibit impairment of bone growth and have shorter body stature. We pinpoint the function of WRN to its helicase domain. We identify short-stature homeobox (SHOX) as a crucial and direct target of WRN and find that the WRN helicase core regulates the transcriptional expression of SHOX via unwinding G-quadruplexes. Consistent with this, *shox*⁻/⁻ zebrafish exhibit impaired bone growth, while genetic overexpression of *SHOX* or *shox* expression rescues the bone developmental deficiency induced in *WRN/wrn*-null mutants both in vitro and in vivo. Collectively, we have identified a previously unknown function of WRN in regulating bone development and growth through the transcriptional regulation of SHOX via the WRN helicase domain, thus illuminating a possible approach for new therapeutic strategies.

Werner Syndrome (WS) is a rare autosomal recessive disorder that exhibits several clinical features associated with accelerated aging[1–3]. It is one of the few adult-onset progerias in which patients usually develop normally until they reach adulthood, except for body stature[3,4]. WS patients not only show grey hair, wrinkles, skin atrophy, and bilateral cataracts, but also show several metabolic dysfunctions – including loss of subcutaneous fat, type 2 diabetes mellitus, hypogonadism, and atherosclerosis – in their late 20 s or early 30 s[5,6]. The first clinical sign, usually regarded retrospectively, is an absence of the growth spurt and a fairly short stature when they reach adolescence[7]. A Japanese clinical report showed that the average body height of WS

patients (around 150–152 cm) was much shorter than that of healthy controls (around 162–164 cm)[8]. Extensive biochemical studies of the pathologies of WS have been focused mainly on DNA damage[1,2], genomic instability[2], and mitochondrial homeostasis[9]. Patients with WS or those with cells missing WRN are highly susceptible to DNA-damaging agents, highlighting the critical involvement of WRN in DNA repair[10]. However, the mechanism behind the short stature in WS patients remains elusive.

Human stature is controlled by the elongation of the bones. Bone formation during embryogenesis usually includes endochondral ossification and intramembranous ossification[11]. Different skeletal

[1]School of Biomedical Sciences, Faculty of Medicine, the Chinese University of Hong Kong, Shatin, N.T., Hong Kong SAR. [2]CUHK-SDU University Joint Laboratory on Reproductive Genetics, the Chinese University of Hong Kong, Shatin, N.T., Hong Kong SAR. [3]Department of Clinical Molecular Biology, University of Oslo and Akershus University Hospital, 1478 Lørenskog, Norway. [4]Hong Kong Branch CAS Center of Excellence for Animal Evolution and Genetics, the Chinese University of Hong Kong, Shatin, N.T., Hong Kong SAR. [5]CAS Key Laboratory of Tissue Microenvironment and Tumor, Shanghai Institute of Nutrition and Health, University of Chinese Academy of Sciences, Chinese Academy of Sciences, Shanghai, China. [6]Department of Biomedical Engineering, the Chinese University of Hong Kong, Shatin, N.T., Hong Kong SAR. ✉e-mail: chanwy@cuhk.edu.hk

compartments develop through specific processes. For example, intramembranous ossification is indispensable in the development of bones such as the skull, facial bones, and pelvis that arise through the direct differentiation of mesenchymal precursors into osteoblasts and then into osteocytes[11,12]. Bones of the axial and appendicular skeleton are generally formed through endochondral ossification, in which an initial cartilage template is systematically replaced by bone[13]. Growth and development of the axial and appendicular bones are complicated and highly organized processes modulated by diverse signaling pathways, in which chondrogenesis is a crucial step[14]. Chondrocytes, which are derived from the undifferentiated mesenchymal lineage in cell condensations, orchestrate the growth and development of axial bone elements[11,13,14]. Defects in cartilage development or chondrogenesis can cause growth retardation, dwarfism, and other severe conditions. However, the potential mechanism of action of WRN in human bone development, especially chondrogenesis, remains a mystery.

Mutations in the *WRN* gene, which is presently the only gene linked to WS, cause the disease[4]. The *WRN* locus is on chromosome 8p12 and is composed of 34 coding exons spanning 140 kb[15]. WRN belongs to the RECQ family, which also includes RECQ1, BLM (RECQ2), WRN (RECQ3), RECQ4, and RECQ5[16,17]. WRN has four domains, one of which is a $3' \rightarrow 5'$ exonuclease domain in its N-terminal area, which makes WRN unique compared to the other RECQ family members[18]. The WRN helicase core consists of two parts: an ATPase domain, which works as a "power ATM" to provide energy, and a $3' \rightarrow 5'$ helicase domain, which is mostly responsible for DNA transactions, such as DNA repair, transcription, and recombination[18,19]. DNA and RNA helicases use the energy released by nucleotide triphosphate hydrolysis to influence nucleic acid structures[20]. WRN helicases are involved in practically all aspects of DNA metabolism and show a preference for binding and unwinding various "bubble" structures, such as triplexes and G4s, in the genome[21,22]. Interestingly, it has also been reported that the G-quadruplexes, which are generally understood to be gene silencers in the genome, are important in regulating gene transcription[23]. Whether WRN is modulating chondrogenesis through its helicase activity and thereby affecting bone growth is so far unstudied.

In this study, we use zebrafish as an in vivo model and discover that *wrn* is indispensable for cartilage development. *Wrn* deficiency results in the inhibition of bone growth and short stature in vivo, highlighting the role of the WRN helicase in the regulation of bone development and growth. Furthermore, through RNA-sequencing (RNA-seq) and chromatin immunoprecipitation sequencing (ChIP-seq), we are able to show that the *SHOX* (short-stature homeobox) gene, which is inextricably linked to human height determination, is a direct target of *WRN* in bone homeostasis. The promoter of *SHOX* has been shown to be rich in guanine through the use of G4 QGRS prediction software. We found that WRN regulates *SHOX* expression through opening these G4 structures and facilitating transcription, demonstrating a process by which WRN dysfunction may cause short stature.

## Results

### The *wrn*$^{-/-}$ mutant zebrafish exhibit shortened body length

To determine the physiological involvement of the endogenous WRN gene in bone homeostasis, we chose zebrafish (*Danio rerio*) as the study model. The fact that WS mice do not greatly mimic the human phenotype is widely established[24]. This could be due to the long telomeres in mice that reduce the need for DNA repair and telomere maintenance[24]. Zebrafish have many advantages over other model systems, including their highly similar skeletal structure with that of humans, transparency for direct observation, and easy manipulation, making them an ideal model to study bone development[25,26]. The *wrn* mutant zebrafish (wrn$^{sa34829}$, C > T) was obtained from ZIRC. After intercrossing and genotyping, the *wrn*$^{-/-}$ zebrafish were successfully generated (Supplementary Fig. 1a, b).

As early as 40 h post-fertilization (hpf), the total body lengths of the *wrn*$^{-/-}$ and wildtype (WT) siblings were measured, and the results showed that the body length of *wrn*$^{-/-}$ was significantly reduced (Fig. 1a). The total body length for the *wrn*$^{-/-}$ and WT siblings was measured at different embryonic stages (4 days post fertilization (dpf) to 14 dpf). The results showed that the *wrn*$^{-/-}$ mutant zebrafish had decreased growth rate (Fig. 1b). Consistently, the mutants exhibited a significant decrease in the total body length compared to the WT groups at 40 dpf. (Supplementary Fig. 1c, d). These data suggest the remarkable growth retardation in the *wrn*$^{-/-}$ zebrafish, highlighting the applicability of this model for WS-related studies.

To analyse the early bone development in the *wrn*$^{-/-}$ embryos and their WT siblings, we performed calcein ($C_{30}H_{26}N_2O_{13}$) staining, which labels bone structures and is useful for studying bone growth[27]. The visualization of the axial skeletal structures, which developed in a progressive fashion from head to tail, began by 7 dpf (Fig. 1c, d). Four to five segments of calcein-stained vertebrae were observed in WT zebrafish, while only two calcein-stained vertebrate columns were observed in *wrn*$^{-/-}$ zebrafish (Fig. 1d). Embryonic skeletal developmental retardation in the *wrn*$^{-/-}$ mutant was exacerbated from 10 dpf to 14 dpf (Fig. 1e, f).

### Loss of *wrn* causes bone abnormalities in zebrafish

The expression profile of *wrn* was evaluated using whole-mount in situ hybridization (WISH) at different stages. In the WT zebrafish, *wrn* was expressed in the cartilage and vertebrae by 7 dpf and 10 dpf (Fig. 2a, b). As expected, the *wrn*$^{-/-}$ mutant zebrafish did not express *wrn*. We then performed masson's trichrome staining to further investigate the underlying changes in bone formation histologically. By 14 dpf, notochord structures were noted and appeared blue along the notochord sheath in WT zebrafish, indicating that the cartilage and extracellular matrix containing collagen II and X had been formed (Fig. 2c). The notochord, which is an essential element in vertebrate development, has two main structures, namely the notochord sheath (nsh) and notochord vaculated cells (vc)[28–31]. However, when analyzing the sections from *wrn*$^{-/-}$ mutants, we failed to detect normal notochord structures but instead noted numerous disorganized unidentifiable structures; the usual blue staining signal was not detected, suggesting a lack of cartilage and extracellular matrix formation (Fig. 2e). By 40 dpf, the WT zebrafish showed the basic elements that are crucial for bone development and growth: notochord sheath (nsh), intercentral joint (ij) and centrum (ct)[29,30,32] (Fig. 2g). These structures were not detectable in *wrn*$^{-/-}$ mutant siblings (Fig. 2i). By 14 dpf and 40 dpf, fluorescence in situ hybridization (FISH) for *wrn* (yellow) combined with bromodeoxyuridine/5-bromo-2'-deoxyuridine (BrdU) staining (green) showed the extensive expression of *wrn* and BrdU in the notochord sheath, indicating that most chondrocytes were displaying a proliferative state in the WT zebrafish (Fig. 2d and h). In contrast, the expression of *wrn* and BrdU were barely detectable in *wrn*$^{-/-}$ mutant zebrafish at both 14 and 40 dpf. These structures were not detectable in *wrn*$^{-/-}$ mutant siblings (Fig. 2f, j). Collectively, these data suggest *wrn* is important for normal bone development.

As early as 7dpf, WISH analysis for *sox9a*, *col2a1a*, and *col10a1a* (classical hypertrophic chondrocyte markers) as well as *col1a1a*, which is expressed in precursors of osteocytes, showed diminished expression in *wrn*$^{-/-}$ mutant zebrafish compared to that in WT (Fig. 2k–n) zebrafish. By 14 dpf, FISH analysis showed the abundant expression of the chondrogenic markers *sox9a* (green), *col2a1a* (red), and *col10a1a* (yellow) in WT embryos (Fig. 2o). In contrast, a lower expression level of *col2a1a* and no expression of *sox9a1a* or *col10a1a* was observed in the *wrn*$^{-/-}$ siblings (Fig. 2q). Extensive BrdU signals were detected in WT fish by 14 dpf (Fig. 2p), whereas only a weak BrdU signal was observed in *wrn*$^{-/-}$ fish (Fig. 2r). By 40 dpf, high expression levels of *col2a1a* (red), *col1a1a* (green), and *col10a1a* (yellow) was observed via FISH analysis in the WT siblings (Fig. 2s), while the expression of these markers was

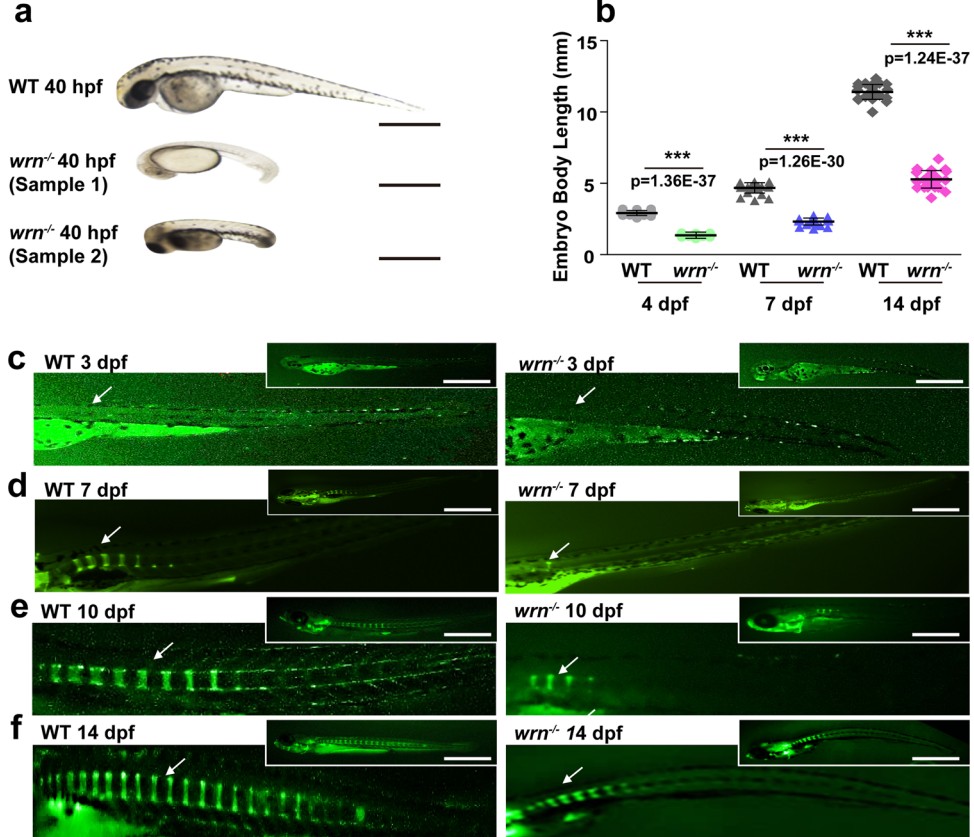

**Fig. 1 | The *wrn* mutant zebrafish exhibit shortened body length.**
**a** Representative bright-field images of 3 independent experiments between WT and *wrn*⁻/⁻ mutant zebrafish at 40 hpf. Scale bar = 100 μm. **b** Dot graph analysis of the total body length of WT and *wrn*⁻/⁻ mutants from 4 dpf to 14 dpf. *N* = 25 independent embryos for WT and *wrn*⁻/⁻ mutants, respectively. Each dot represents a biological replicate. **c–f** Representative calcein green staining of 3 independent experiments to examine bone formation between WT and *wrn*⁻/⁻ mutant zebrafish from 3 dpf to 14 dpf. White arrows indicate vertebrate regions. Scale bar = 100 μm. Data are presented as the mean ± S.D. Statistical analysis was performed using two-tailed unpaired Student's *t*-test. *$P < 0.05$, **$P < 0.01$, ***$P < 0.001$.

only faintly observed in *wrn*⁻/⁻ fish (Fig. 2u). Moreover, BrdU staining analysis showed weaker signals in WT fish at 40 dpf compared to 14 dpf (Fig. 2t), indicating that the chondrocytes in WT siblings might be at prehypertrophic or hypertrophic stages, which are characterized by chondrocyte apoptosis and the expression of the osteocyte precursor markers such as *col1a1a*[33]. In addition, we observed a weak BrdU signal in the *wrn*⁻/⁻ siblings by 40 dpf (Fig. 2v). To exclude the possibility that chondrocytes were negatively selected by apoptosis, in situ apoptosis assay was performed. As shown in Supplementary Fig. 2a, a larger number of apoptosis signals were found in *wrn*⁻/⁻ zebrafish compared with that in the wildtype in the vertebrate regions. We also found an increased expression of γH2AX in *wrn*⁻/⁻ zebrafish, indicating that early apoptosis could be attributed to the accumulation of DNA damage (Supplementary Fig. 2b). Taken together, these data demonstrate that *wrn* deficiency caused bone developmental deficiency in zebrafish.

### *WRN* deficiency impairs cartilage development in vitro

Given the observed stunted growth phenotype and the impairment of bone development in *wrn*⁻/⁻ mutant fish, we next investigated the potential specific stages of bone development when *WRN* functions. It has been reported that chondrogenesis is a crucial and early step required for the elongation of the bones and an increase in body height[34]. Two stem cell models were used: human embryonic stem cells (hESCs), and human mesenchymal stem cells (hMSCs), to better study chondrogenesis in Werner Syndrome (Fig. 3a and Supplementary Fig. 3a). Previously reported procedures were used to differentiate hESCs[35] and hMSCs[36]. The gene expression of *WRN* increased gradually during chondrogenic development, with the highest level reached at

day 14 after initiation of differentiation in both wildtype (name as CTR) hESCs and hMSCs, suggesting an important role for *WRN* in chondrogenesis (Fig. 3b and Supplementary Fig. 3b). To further assess the effects of *WRN* on cartilage development and to identify the specific stage of chondrogenesis where the *WRN* gene is active, we generated *WRN* knockdown (name as shWRN1# and shWRN2# respectively) hESCs and hMSCs via lentivirus transfection, and the efficiency of the gene knockdown was measured by qRT-PCR (Fig. 3c and Supplementary Fig. 3c).

An Alcian blue assay and qRT-PCR were performed to investigate the role of WRN in the process of chondrogenesis. In stage 1 (day 0), the CTR, shWRN1#, and shWRN2# cells all retained their hESC morphology (Fig. 3d), and there was no significant difference in the expression of pluripotency markers including – *OCT4* (also known as *POU5F1*), *NANOG*, and *SOX2* – among the CTR, shWRN1#, and shWRN2# cells (Fig. 3e–g). At stage 2 (day 4), the CTR, shWRN1#, and shWRN2# hESCs became more condensed and no difference was noted (Fig. 3h). In addition, although some pluripotency markers, such as *OCT4* and *NANOG*, were continuously expressed, the expression of *CDH1* (also known as *ECAD*) which is associated with the mesendoderm phenotype was increased on day 4. Further, the expression of *GSC* and *MIXL1*, regarded as markers of primitive streak mesendodermal cells, was enhanced rapidly. However, the expression of these three genes showed no statistical differences among the CTR, shWRN1#, and shWRN2# cells (Fig. 3i–k). Together, these data suggest that the cells became a population enriched specifically for mesendodermal growth. It has been reported that the loss of *WRN* did not affect the pluripotency of induced stem cells (iPSCs); however, upon differentiation

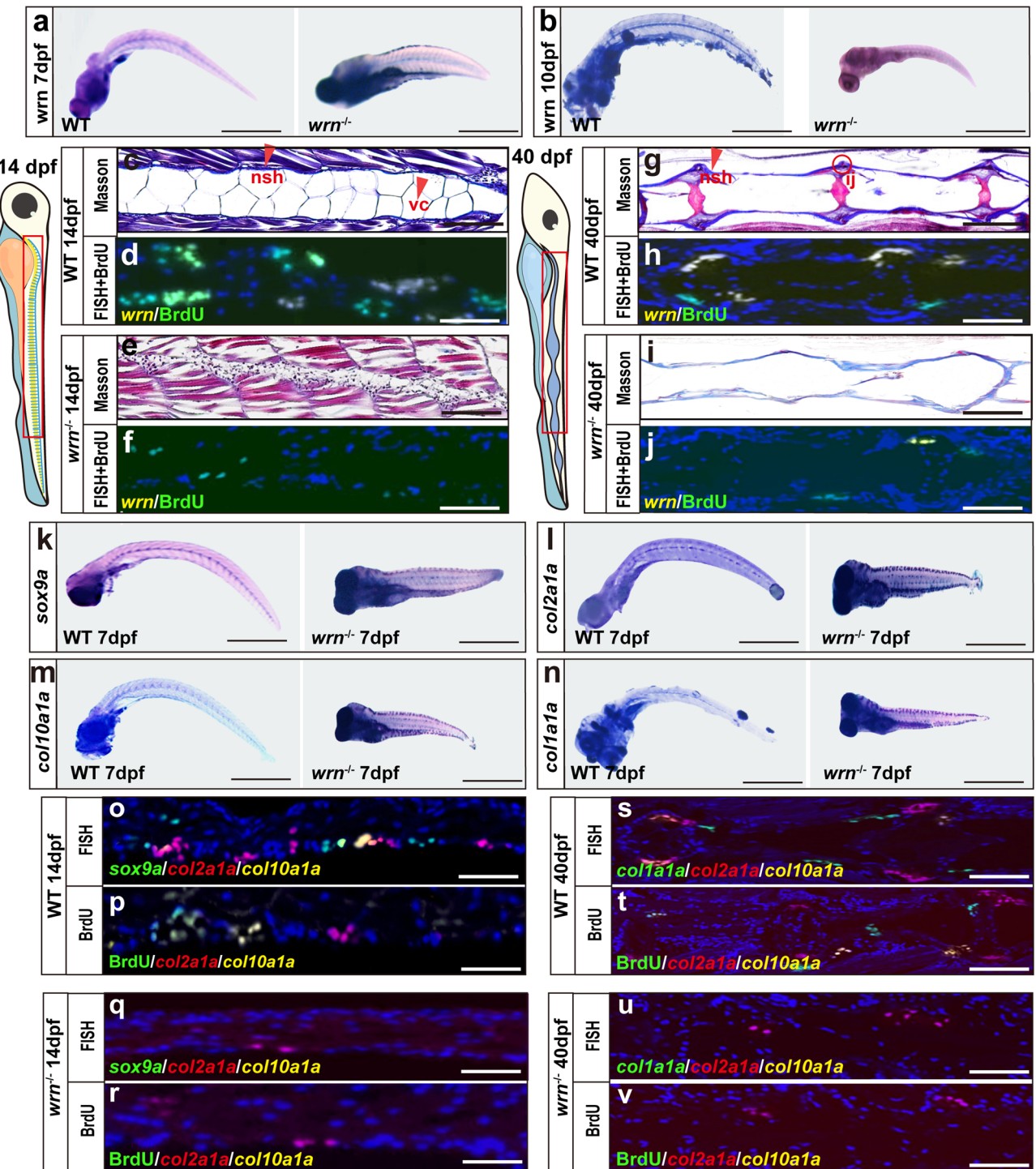

**Fig. 2 | Loss of *wrn* causes bone abnormalities in zebrafish. a, b** Representative WISH analysis of 3 independent experiments of *wrn* expression between WT and *wrn⁻/⁻* mutant zebrafish on both 7 dpf and 10 dpf. **c, e, g, i** Representative Masson's trichrome staining of 3 independent experiments between WT and *wrn⁻/⁻* mutant zebrafish on 14 dpf (**c** WT; **e** *wrn⁻/⁻* mutants) and 40 dpf (**g** WT; i, *wrn⁻/⁻* mutants). Scale bar = 100 μm. Notochord sheath (nsh, red arrow indicated) and notochord vaculated cells (vc, red arrow indicated). Intercentral joint (ij, red encircled) and centrum (ct). **d, h, f, j** Representative FISH analysis of 3 independent experiments of *wrn* expression and BrdU staining between WT and *wrn⁻/⁻* mutant zebrafish on

14 dpf (**d** WT; **f** *wrn⁻/⁻* mutants) and 40 dpf (**h** WT; **j** *wrn⁻/⁻* mutants). **k–n** Representative WISH analysis of 3 independent experiments of chondrogenic markers (*sox9a, col2a1a, col10a1a,* and *col1a1a*) between WT and *wrn⁻/⁻* mutant zebrafish on 7 dpf. Scale bar = 50 μm. **o–v** Representative FISH analysis combined with BrdU staining of 3 independent experiments of chondrogenic markers (*sox9a, col2a1a,* and *col10a1a*) between WT and *wrn⁻/⁻* mutant zebrafish on 14 dpf (**o, p** WT; **q, r** *wrn⁻/⁻* mutants) and chondrogenic markers (*col2a1a, col10a1a,* and *col1a1a*) on 40 dpf (**s, t** WT; **u, v** *wrn⁻/⁻* mutants). Scale bar = 100 μm.

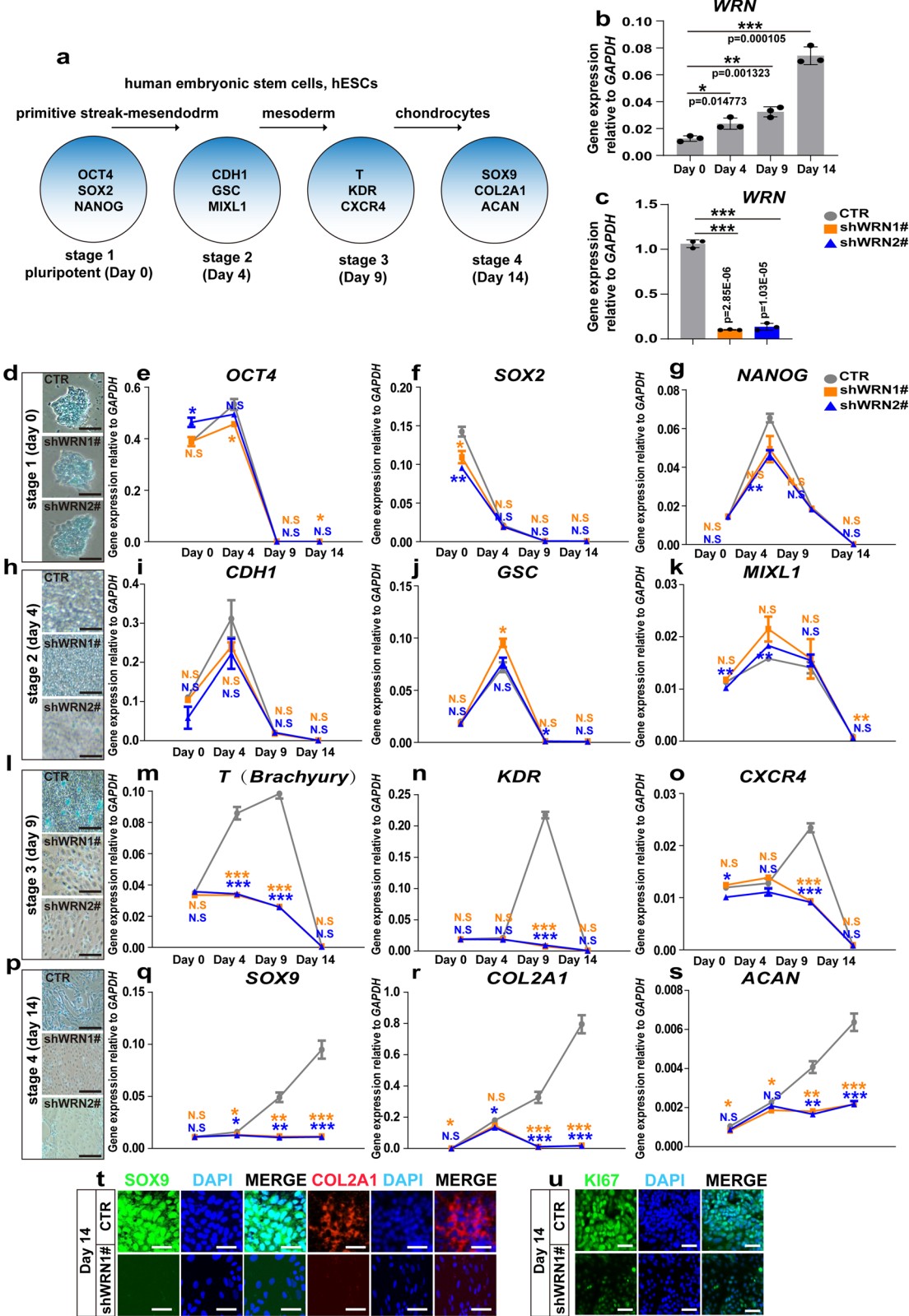

into the mesenchymal lineage, the differentiation ability of *WRN*[−/−] iPSCs was influenced[37]. We hereby continued to examine the differentiation at stage 3 where the CTR cell clusters were distributed throughout the culture and stained in dark blue, indicating the existence of an extracellular matrix, which is a necessary prerequisite for chondrocyte formation (Fig. 3l). However, shWRN1# and shWRN2# cells were not packed as tightly as the control cells. The early

mesoderm markers, such as *T* (also known as *BRACHURY*), *KDR*, and *CXCR*4, were increased in the CTR cells but were decreased in the shWRN1# and shWRN2# cells (Fig. 3m–o). At stage 4 (day 14) of chondrocyte development, the CTR cells showed a rounded chondrocyte-like morphology, while shWRN1# and shWRN2# cells did not exhibit the same phenotype (Fig. 3p). The expression levels of chondrocyte markers, including *SOX9*, *COL2A1*, and *ACAN* were all

**Fig. 3 | WRN deficiency impairs cartilage development in vitro. a** Illustration of hESCs differentiation model. **b** qRT-PCR measurement of *WRN* expression profile during chondrogenesis in hESCs. *N* = 3 independent biological experiments. **c.** qRT-PCR measurement of WRN-KD efficiency. *N* = 3 independent biological experiments. **d, h, l, p** Representative alcian blue staining of 3 independent experiments to examine the formation of chondrocyte on day 0 (**d**), day 4 (**h**), day 9 (**l**), and day 14 (**p**) between CTR and two KD-*WRN* groups. Scale bar = 20 μm. **e−g** qRT-PCR measurement of genes (*NANOG*, *OCT4*, and *SOX2*) related to hESC pluripotency stage. *N* = 3 independent biological experiments. **i−k** qRT-PCR measurement of genes (*MIXL1*, *GSC*, and *CDH1*) related to the primitive streak-mesendodermal stage. *N* = 3 independent biological experiments. **m−o.** qRT-PCR

measurement of genes (*T*, *KDR*, and *CXCR4*) related to the mesodermal stage. *N* = 3 independent biological experiments. **q−s** qRT-PCR measurement of genes (*SOX9*, *COL2A1*, and *ACAN*) related to chondrocytes. *N* = 3 independent biological experiments. **t** Representative immunofluorescent staining of 3 independent experiments between the CTR and KD-*WRN* groups on day 14 in hESCs. SOX9 and COL2A1 were examined. Scale bar = 50 μm. **u** Representative images of immuno-fluorescent staining of 3 independent experiments between the CTR and KD-*WRN* groups on day 14 in hESCs. Ki67 was examined. Scale bar = 20 μm. Data are presented as the mean ± S.D. Statistical analysis was performed using two-tailed unpaired Student's *t*-test. \*P < 0.05, \*\*P < 0.01, \*\*\*P < 0.001.

significantly inhibited in the shWRN1# and shWRN2# cells compared to the CTR cells (Fig. 3q−s).

hMCSs have been reported to undergo an intrinsic differentiation program that is reminiscent of chondrocytes, especially the process of endochondral bone formation which is crucial for chondrogenesis and bone growth[36]. The expression of *KDR* and *T* decreased in the shWRN1# and shWRN2# hMSCs compared to the CTR cells, confirming the results in hESCs that depletion of *WRN* caused the mesenchymal lineage differentiation dysregulation (Supplementary Fig. 3d-e). Intriguingly, the expression of *SOX9*, *COL2A1*, *SOX6*, *COL10A1*, *ACAN*, and *MMP10* was all significantly downregulated in the shWRN1# and shWRN2# hMSCs compared to the CTR hMSCs (Supplementary Fig. 3f−k).

A significant reduction of SOX9 and COL2A1 was observed when WRN was depleted (Fig. 3t and Supplementary Fig. 3l). Additionally, immunofluorescence for Ki67 showed an abundant expression in the CTR cells but not in the shWRN1# and shWRN2# hESCs or hMSCs at day 14 of differentiation (Fig. 3u and Supplementary Fig. 3m). In addition, increased apoptotic cells (Supplementary Fig. 4a, b) and the upregulation of *BCL2* and *CASPASE8* in both shWRN1# and shWRN2# hESCs and hMSCs were noted at day 14 of differentiation (Supplementary Fig. 4c−f), this was similar to findings observed in the zebrafish model (Supplementary Fig. 2a). Collectively, our data consistently suggest that WRN deficiency impairs cartilage development in hESCs and hMSCs.

## The WRN helicase is essential for chondrogenesis
As a helicase, WRN contains an exonuclease domain, an ATPase domain, a RecQ helicase domain, and a helicase-and-ribonuclease D/C-terminal (HRDC) domain[21]. To better confirm the specific role of WRN in chondrogenesis, the expression of Bloom Syndrome protein (BLM, the other helicase protein) was also examined both in vitro and in vivo. As shown in Supplementary Fig. 5a−c, no significant difference was noted between the wildtype and WRN/*wrn*-KD groups. These results indicate that WRN is specific to chondrogenesis in WS. Next, to investigate how WRN regulates chondrogenesis, four WRN mutant lentiviruses, namely X-WRN (E84A, exonuclease dead), K-WRN (K577M, ATPase dead), R-WRN (R993A, helicase dead), and F-WRN (F1037A, helicase dead) were created following previous report[38]. Both the WRN-KD hESCs and WRN-KD hMSCs (shWRN1# cell lines were used) were infected with lentivirus generating cell lines expressing either full-length or mutated *WRN*. (The cells infected with the blank vector were used as the control, names as vector). Full-length *WRN* stimulated the expression of *SOX9* and *COL2A1* more significantly in comparison with that in the vector group on day 14, confirming the importance of *WRN* in chondrogenesis (Fig. 4a, b). X-WRN group also stimulated the expression of *SOX9* and *COL2A1* slightly, but showed a lower level of expression compared to full-length *WRN* group, indicating that exo-nuclease activity was partial crucial for chondrogenesis. However, the K-WRN, R-WRN, and F-WRN helicase mutants all failed to induce the expression of *SOX9* and *COL2A1*, and similar findings were confirmed in the hMSC model (Fig. 4c, d), suggesting the importance of the WRN

helicase in chondrogenesis. Immunofluorescent staining of SOX9 and COL2A1 showed the same results (Fig. 4e).

The in vitro stem cell findings were validated in zebrafish by microinjecting the different human *WRN* mRNAs into the *wrn*−/− mutant zebrafish at the one-cell stage. No strong calcein green signal was detected in the helicase mutants (K-*WRN*, R-*WRN*, and F-*WRN*) at 7 dpf (Fig. 4f), and no expression of *sox9a* and *col2a1a* was observed at 7 dpf by using WISH analysis (Fig. 4g, h). However, X-WRN slightly stimulated the expression of *sox9a* and *col2a1a*, confirming that the exo-nuclease activity was not essential for chondrogenesis. Together, our findings in the hESCs, hMSCs, and zebrafish strongly support an indispensable role for the WRN helicase in chondrogenesis.

## Integrative analysis of RNA-seq and ChIP-seq in chondrogenesis
To identify *WRN*-mediated transcriptional targets that might account for short stature in WS, RNA-seq analysis was performed to compare the transcriptomes of WRN (shWRN1#) and the CTR hESCs at four sequential timepoints after initiating chondrogenesis (day 0, day 4, day 9, and day 14). A gene-wide hierarchical clustering heatmap was produced to examine all differentially expressed genes (Supplementary Fig. 6a). Gene ontology (GO) analysis at the four stages showed that chondrocyte differentiation, cartilage development, and helicase activity decreased at day 14 in WRN hESCs (shWRN1#) compared with that in the CTR hESCs (Supplementary Fig. 6b−e). Furthermore, vol-cano plots showed numbers of significantly downregulated (red, WRN (shWRN1#) vs CTR on day 14) and upregulated genes (blue, WRN (shWRN1#) vs CTR on day 14) (Fig. 5a). Among these genes like *SHOX* and *COL9A3*, which are crucial components of cartilage[39,40], were dra-matically reduced in WRN cells vs. CTR cells. Thus, our RNA-seq data were in line with the aforementioned cellular and tissue data, pointing to the importance of WRN in chondrogenesis.

Results of the intersection of ChIP-seq and RNA-seq (Fig. 5b) showed that 380 genes were directly regulated by *WRN* in chondrocyte homeostasis: 116 upregulated and 264 downregulated. Cartilage development, cell growth, and regulation of cell adhesion, which are crucial processes in chondrogenesis[12,13,41], were all downregulated in WRN hESCs (shWRN1#). GO analysis of targets directly modulated by *WRN* demonstrated that these pathways were significantly down-regulated WRN hESCs (shWRN1#) (Fig. 5c).

In-depth heatmap analysis of representative genes clustered in the genes involved in bone development, cell growth, and spinal cord development were further performed. The significantly changed genes (WRN (shWRN1#) vs. CTR) in each group were presented in Supple-mentary Fig. 6f-g. Differentially altered potential direct targets were validated using qRT-PCR in both the hESCs and hMSCs on day 14 during chondrogenesis. Among many genes displaying changes, the short-stature homeobox gene (*SHOX*) was the most significantly downregulated and the same pattern was seen in both hESCs and hMSCs (Fig. 5d, e). Then the expression patterns of *wrn* and *shox* were evaluated at 40 dpf (most of the zebrafish tissues and organs have formed) both in wildtype and *wrn*−/− zebrafish. The *wrn* gene was found to be universally expressed in the wildtype zebrafish, and the *shox*

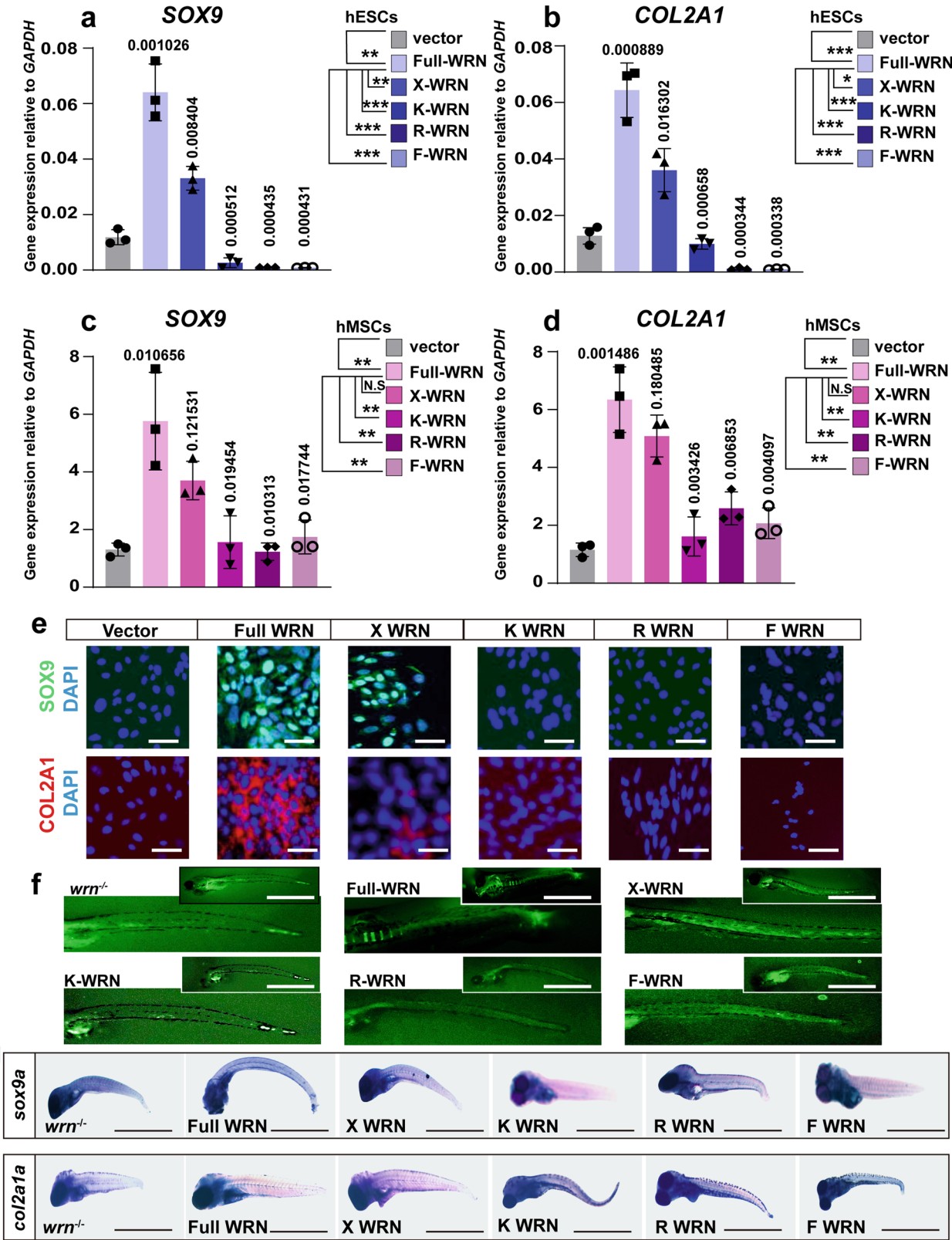

**Fig. 4 | WRN helicase is essential for chondrogenesis. a–d.** qRT-PCR measurement of different *WRN* expressions on day 14 in hESCs (**a, b**) and hMSCs (**c, d**). (Vector, full-length *WRN*, X-*WRN* (E84A), K-*WRN* (K57M), R-*WRN* (R993A), F-*WRN* (F1037A)). *N* = 3 biological independent experiments. Data are presented as the mean ± S.D. Statistical analysis was performed using two-tailed unpaired Student's *t*-test. *P < 0.05, **P < 0.01, ***P < 0.001. **e** Representative immunofluorescent staining of 3 independent experiments between the CTR and KD-*WRN* group on day 14 in hESCs. SOX9 and COL2A1 were examined. Scale bar = 50 μm. **f** Representative calcein staining of three independent experiments to examine the bone formation after microinjection of different human *WRN* mRNA in one-cell stage and checked at 7 dpf. Scale bar = 100 μm. **g, h** Representative WISH analysis of 3 independent experiments to examine the expression of *sox9a* and *col2a1a* after microinjection of different human *WRN* mRNA in one-cell stage and checked on 7 dpf. Scale bar = 50 μm.

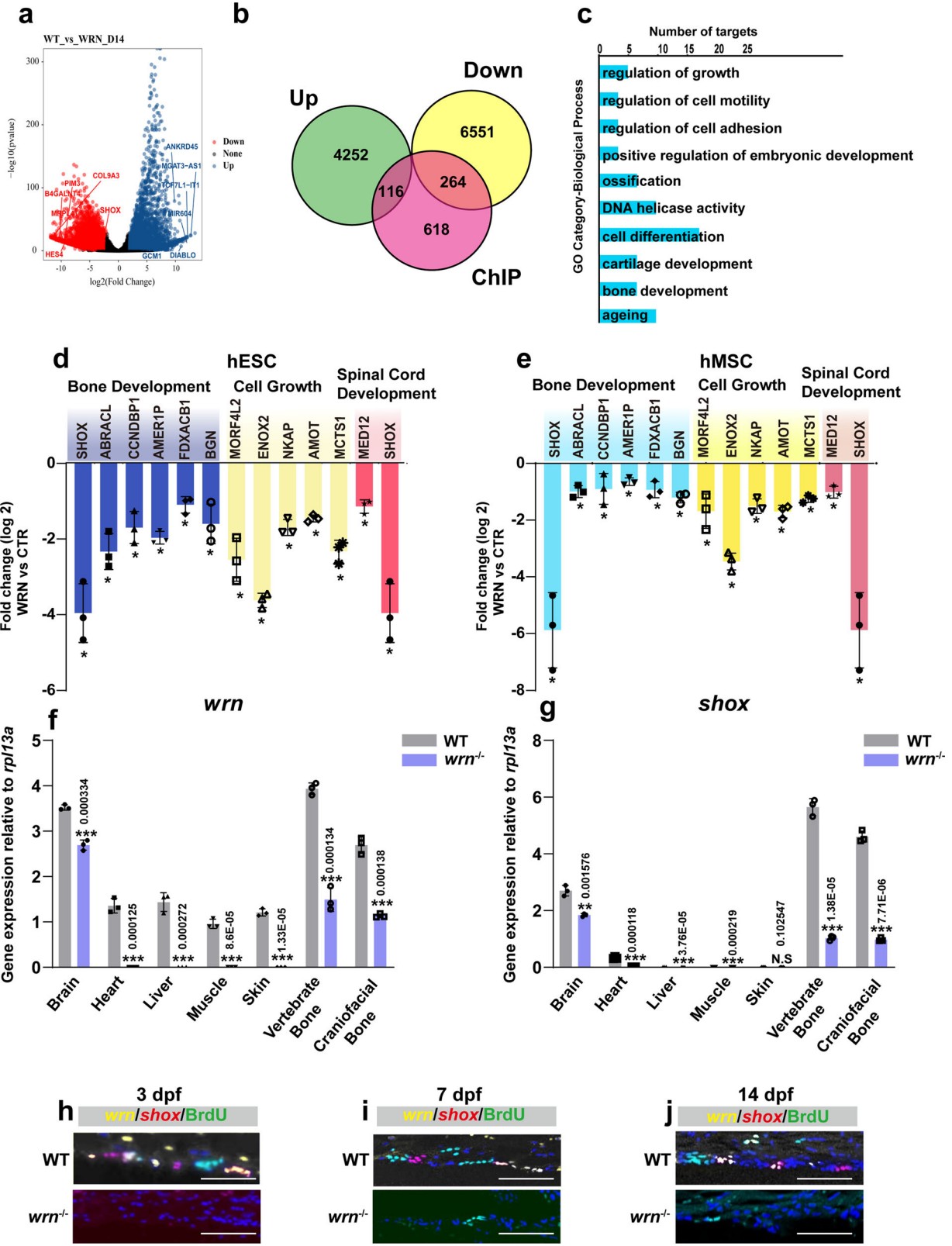

gene mainly expressed in the bones (the vertebrate bones and craniofacial bones), also slightly in the brain, indicating that WRN-SHOX interaction would be crucial in bone metabolism (Fig. 5f, g). As Fig. 5h−j shown, *shox* (red), *wrn* (yellow), *and* BrdU (green) signals accumulated in wildtype zebrafish from 3 to 14 dpf according to FISH analysis; this accumulation was not observed in *wrn*[−/−] zebrafish, indicating that *wrn* and *shox* were co-expressed during chondrogenesis. Combined, the

in vitro and in vivo data indicate that SHOX might be the downstream target of WRN in chondrogenesis.

## SHOX is a crucial regulator in WS bone development

To determine the role of *SHOX* in bone development, we examined the normal expression profile of *SHOX* in hESCs and hMSCs. *SHOX* expression was first observed in the mesoderm stage (day 9) and

**Fig. 5 | Integrative analysis of RNA-seq and ChIP-seq in chondrogenesis.**
**a** Volcano plot was depicted with the fold change on day 14. *P* values were calculated by empirical Bayes moderated *t* test in limma combined with Benjamini–Hochberg method adjustment. Genes with an adjusted *p*-value of ≤ 0.05 and |log2 fold change| > 0.5 were considered as significantly changed. Top upregulated genes (blue) and downregulated changed genes (red) were shown. **b** The overlay of RNA-seq and ChIP-seq analysis showed the genes were potential direct targets of *WRN* in chondrogenic homeostasis. **c** GO functional clustering of genes that were for identification of biological processes directly regulated by *WRN*. Representative downregulated categories were shown. **d, e** qRT-PCR validation analysis showed the mRNA expression fold change among bone development, cell

growth, and spinal cord development-associated genes in the CTR vs *WRN*-KD chondrocytes in the hESCs (**g**) and hMSCs (**h**) on day 14. *N* = 3 independent biological experiments. The error bar represents the standard deviation (s.d.) and *P*-value was generated by using one-way ANOVA with the Tukey's post hoc test. **f, g** qRT-PCR measurement of the expression of *wrn* and *shox* in different tissue in WT and *wrn*⁻/⁻ mutant zebrafish at 40 dpf. *N* = 3 independent biological experiments. Data are presented as the mean ± S.D. Statistical analysis was performed using two-tailed unpaired Student's *t*-test. **h–j** Representative FISH analysis of three independent experiments of *wrn* and *shox* in WT zebrafish and *wrn*⁻/⁻ mutant zebrafish at 3 dpf, 7 dpf, and 14 dpf. Scale bar = 100 μm. *P < 0.05, **P < 0.01, ***P < 0.001.

increased from then on both in hESCs (Fig. 6a and Supplementary Fig. 7a) and hMSCs (Supplementary Fig. 8a–b), suggesting that WRN and SHOX might coordinate in the regulation of chondrogenesis.

The specific stage when SHOX gene affects chondrogenesis was investigated through the use of the *SHOX* knockdown (name as shSHOX1# and shSHOX2#) hESCs and hMSCs with lentivirus (Supplementary Fig. 7b and Supplementary Fig. 8c). Data from Alcian blue staining and qRT-PCR showed an impairment of chondrogenesis in shSHOX1# and shSHOX2# hESCs similar to that of shWRN1# and shWRN2# hESCs (Figs. 3 and 6b). There were no significant differences among the pluripotency markers or primitive streak mesendodermal markers at stage 1 (*OCT4*, *SOX2*, and *NANOG*) (Supplementary Fig. 7c–e) and stage 2 (*CDH1*, *GSC*, and *MIXL1*) (Supplementary Fig. 7f–h) in among the CTR, shSHOX1#, and shSHOX2# cells, while the expression of stage 3 (*T*, *KDR*, and *CXCR4*) (Supplementary Fig. 7i–k) and stage 4 (*SOX9*, *COL2A1*, and *ACAN*) (Supplementary Fig. 7l–n) markers was down-regulated. Consistent with this, immunostaining for SOX9 and COL2A1 was decreased in the shSHOX1# hESCs compared to the CTR hESCs (Fig. 6c). Immunofluorescence for Ki67 showed a reduced expression in the shSHOX1# hESCs at day 14 of chondrocyte differentiation (Fig. 6d). Furthermore, we consolidated our findings by performing similar experiments using hMSCs and obtained similar results (Supplementary Fig. 8d–m). Together, these results indicate an important role for SHOX in chondrogenesis.

To validate our findings in vivo, we again turned to the zebrafish model and used the *shox* mutant zebrafish (*shox*^sa41471^, A > T) (ZIRC) (Supplementary Fig. 9a-b). Both the *shox*⁺/⁻ and *shox*⁻/⁻ mutants showed stunted growth rates compared to their WT siblings by 14 dpf and 40 dpf (Fig. 6e-f). Bone formation in *shox* mutant zebrafish was investigated using calcein green staining which identified that the mineralization region of the vertebral column in *shox*⁺/⁻ and *shox*⁻/⁻ mutants declined in comparison to WT siblings by 7dpf (Fig. 6g). By 10 dpf and 14 dpf respectively, a marked reduction of bone formation was observed in *shox*⁺/⁻ and *shox*⁻/⁻ mutants in comparison with their WT siblings (Fig. 6h-i), indicating that *shox* was vital for bone development and growth.

In WT zebrafish, *shox* was expressed in the skull and somites by 4 dpf and in the axial bones by 7 dpf, and it continued to be expressed in the vertebral column at 10 dpf in WT siblings (Supplementary Fig. 9c, f, i). However, in the *shox*⁺/⁻ and *shox*⁻/⁻ mutant zebrafish, the expression of *shox* was not observed in the vertebrae (Supplementary Fig. 9d, e, g, h, j, and k). Additionally, the expression of *sox9a*, *col10a1a*, *col2a1a*, and *col1a1a* in the vertebrae was decreased in *shox* mutant zebrafish compared to that in WT siblings at 7 dpf using WISH analysis (Fig. 6j-m), indicating that the loss of *shox* blocked the chondrocyte differentiation and bone elongation in zebrafish.

## SHOX restores chondrogenesis in WS
To verify that *SHOX* is the direct downstream effector of *WRN* in regulating stature, *SHOX* was overexpressed in shWRN1# hESCs and shWRN1# hMSCs. The mRNA expression of *SOX9* and *COL2A1* significantly increased compared with that in shWRN1# hESCs and shWRN1# hMSCs on day 14, as shown by qRT-PCR (Fig. 7a, b, d, e) and

immunofluorescence (Fig. 7c, f). More importantly, no significant differences were noted between the CTR cells and rescued ones.

The effect of *SHOX* expression on stature was examined by microinjection of human *SHOX* mRNA into zebrafish embryos at the one-cell stage. By 14 dpf, the total body length normalized by *SHOX* overexpression compared with *wrn*⁻/⁻ mutants (Fig. 7g, h). Similar findings were noted on 21 dpf (Fig. 7i, j).

FISH analysis was performed at 14 dpf and 21 dpf (Figs. 7l–q and 7s–x), showing that the signal of *sox9a* (green), *col2a1a* (red), and *col10a1a* (yellow) increased compared with that in the *wrn*⁻/⁻ mutants, which was agreed with the qRT-PCR results (Fig. 7k, r). In addition, *col2a1a*, *col10a1a*, and BrdU signals were detected again when *SHOX* was overexpressed in *wrn*⁻/⁻ mutant zebrafish. The chondrocyte biology and regulators are complex and depend on several factors[12]. Other SOX family members hereby were evaluated. As results shown (Supplementary Fig. 10a–d), overexpression of *shox* promoted the expression of *sox9b*, *sox5*, and *sox6* compared with that in *wrn*⁻/⁻ mutant zebrafish as well. Together, these data demonstrate that *SHOX* is crucial for controlling bone growth and development both in vitro and in vivo.

## SHOX/shox prevents chondrocyte senescence in WS models
A previous report showed that stem/progenitor cells in the metaphysis of long bones slowed the rate of proliferation when cells became aged, indicating that cellular senescence is a key mechanism for controlling bone growth[42]. To determine whether loss of WRN leads to the accumulation of senescent chondrocytes, we performed senescence assays both in vitro and in vivo. We first examined cellular senescence using senescence flow cytometry analysis by labeling senescence-associated-β-galactosidase (SA-β-gal) positive cells (P3) (Gating strategy was provided in Supplementary Fig. 11a and b) Representative flow cytometry analysis revealed an increased number of SA-β-gal positive cells, with 29.6% of senescent cells detected in shWRN1# hESCs (Fig. 8b) and 30.1% in shWRN1# hMSCs (Fig. 8e) (The senescent cells of CTR-hESCs and CTR-hMSCs were indicated in Fig. 8a, d). After overexpression of *SHOX* in shWRN1# hESCs and shWRN1# hMSCs, the number of senescent cells decreased to 13.1% and 14.1%, respectively (Fig. 8c, f) (Three biological replicates were provided in Supplementary Fig. 11c, d). The expression of two senescence markers, *P53* and *P16*^Ink4a^, was evaluated as well. As results shown, the mRNA expression of *P53* and *P16*^Ink4a^ increased in shWRN1# hESCs and shWRN1# hMSCs compared to that in CTR cells, and overexpression of *SHOX* decreased the mRNA expression of these two genes (Fig. 8g–j). Similar results were obtained in zebrafish (Fig. 8k–m). Together, these results indicate that loss of WRN causes cellular senescence, and overexpression of *SHOX* prevents the senescence phenotype.

## WRN helicase unwinds *SHOX* G-quadruplexes
Motivated by the observations that SHOX is a direct downstream component in WRN-regulated chondrogenesis, we next questioned how WRN regulated SHOX expression. To explore the underlying molecular mechanisms, the effect of the different WRN mutants on

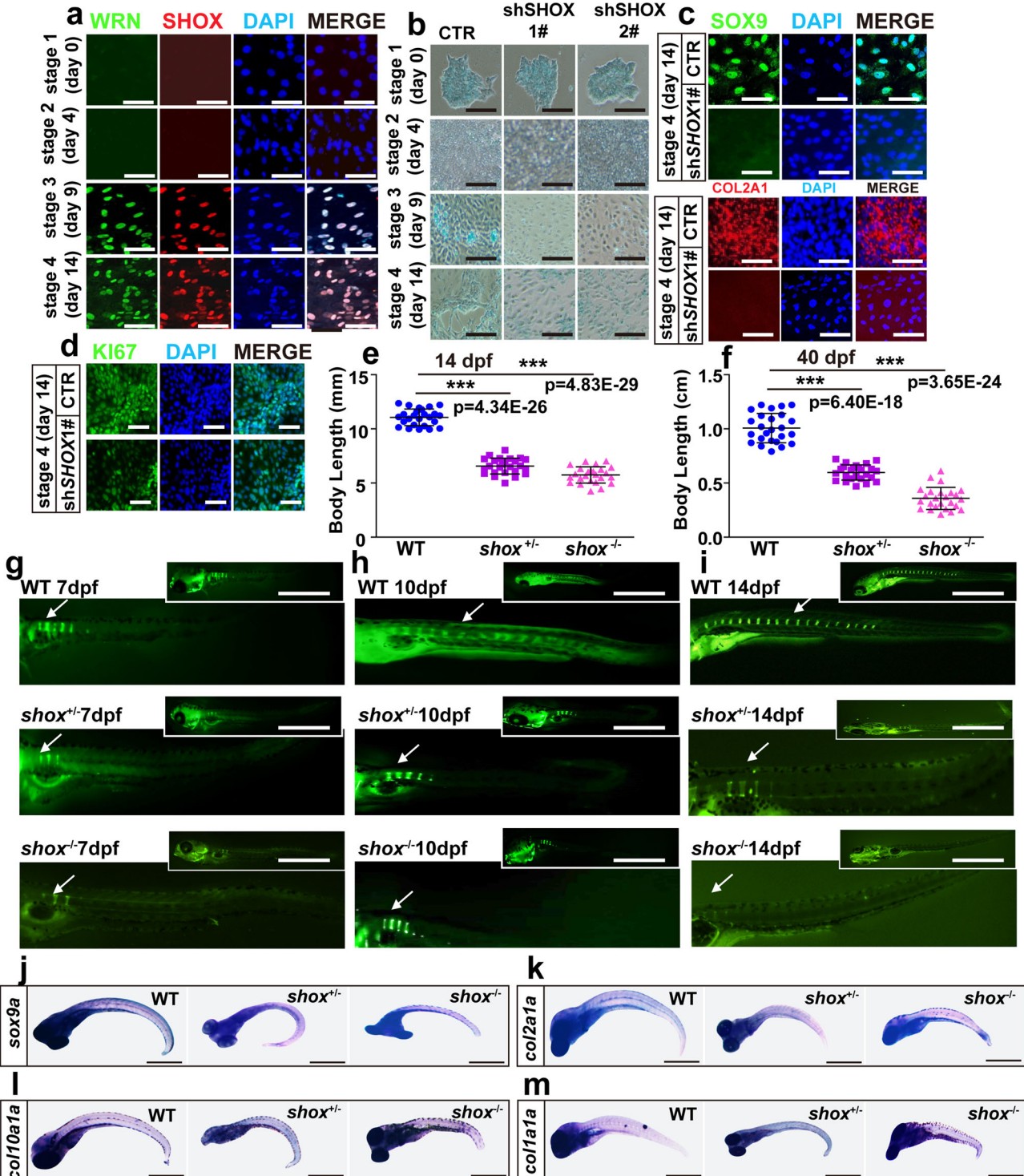

**Fig. 6 | SHOX is a crucial regulator in WS bone development. a** Representative immunofluorescent staining of 3 independent experiments in CTR-hESCs with four sequential time points. WRN and SHOX were examined. Scale bar = 50 μm. **b** Representative alcian blue staining of 3 independent experiments between CTR and two KD-*SHOX* groups. Scale bar = 20 μm. **c** Representative immunofluorescent staining of 3 independent experiments between CTR and shSHOX1# groups on day 14 in hESCs. SOX9 and COL2A1 were examined. Scale bar = 50 μm. **d** Representative immunofluorescent staining of 3 independent experiments between CTR and shSHOX1# groups on day 14 in hESCs. Ki67 was examined. Scale bar = 20 μm. **e, f** Dot graph analysis of the total body length among WT, *shox*[+/-], and *shox*[-/-]

mutants on 14 dpf (**e**) and 40 dpf (**f**). Each dot represents a biological replicate. *N* = 25 independent embryos for WT, *shox*[+/-], and *shox*[-/-] mutants, respectively. Three independent biological experiments were performed. Data are presented as the mean ± S.D. **g–i** Representative calcein staining of 3 independent experiments among WT, *shox*[+/-], and *shox*[-/-] mutants from 7 dpf to 14 dpf. White arrows indicate vertebrate regions. Scale bar = 100 μm. **j–m** Representative WISH analysis of 3 independent experiments of chondrogenic markers (*sox9a, col2a1a, col10a1a,* and *col1a1a*) on 7 dpf. Scale bar = 50 μm. Statistical analysis was performed using two-tailed unpaired Student's *t*-test. *P < 0.05, **P < 0.01, ***P < 0.001.

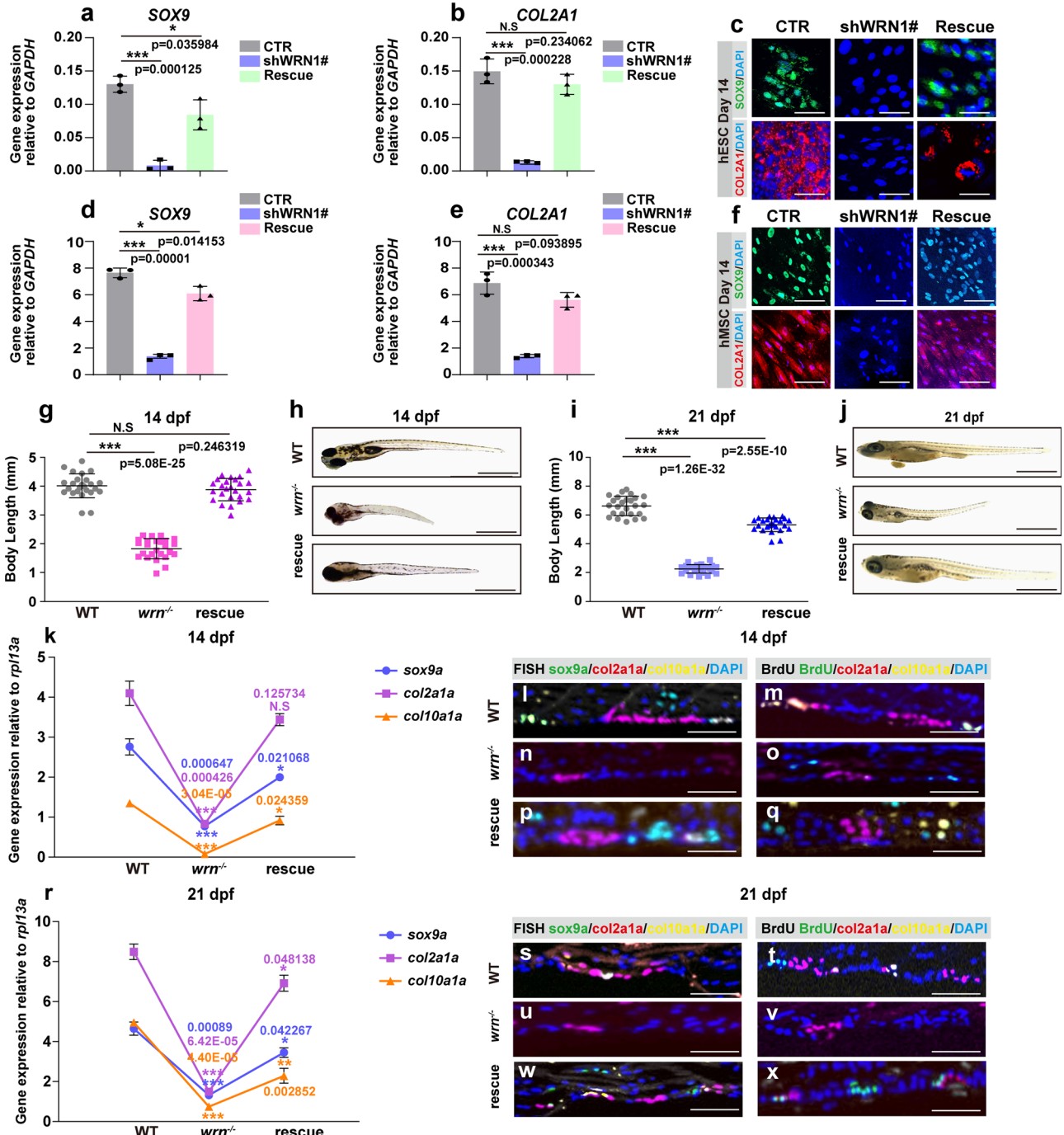

**Fig. 7 | SHOX restores chondrogenesis in WS. a**, **b**, **d**, **e** qRT-PCR measurement of the expression of *SOX9* and *COL2A1* among CTR, shWRN1#, and rescue groups during chondrogenesis on day 14 in hESCs and hMSCs. *N* = 3 independent biological experiments. **c**, **f** Representative immunofluorescent staining of 3 independent experiments among CTR, shWRN1#, and rescue groups on day 14 in hESCs and hMSCs. SOX9 and COL2A1 were examined. Scale bar = 50 μm. **g**, **i** Dot graph analysis of the total body length among WT, *wrn*⁻/⁻ mutant, and rescue groups on 14 dpf (**h**) and 21 dpf (**i**). *N* = 25 independent zebrafish embryos for WT, *wrn*⁻/⁻ mutant, and rescue groups, respectively. Each dot represents a biological replicate.

**h**, **j** Representative bright-field images of 3 independent experiments among WT, *wrn*⁻/⁻ mutant, and rescue groups on 14 dpf (**h**) and 21 dpf (**j**). Scale bar = 100 μm. **k**, **r** qRT-PCR measurement of genes (*sox9a*, *col2a1a*, and *col10a1a*) on 14 dpf (**k**) and 21 dpf (**r**). *N* = 3 independent biological experiments. **l**–**q** and **s**–**x** Representative FISH and BrdU analysis of 3 independent experiments of chondrogenic markers (*sox9a*, *col2a1a*, and *col10a1a*) on 14 dpf (**l**–**q**) and 21 dpf (**s**–**x**). Scale bar = 100 μm. Data are presented as the mean ± S.D. Statistical analysis was performed using two-tailed unpaired Student's *t*-test. *P < 0.05, **P < 0.01, ***P < 0.001.

*SHOX* expression was evaluated. qRT-PCR analysis showed that full length-WRN and X-WRN increased *SHOX* expression compared to the WRN-KD hESCs and WRN-KD hMSCs (shWRN1# cell lines were used). However, the transcriptional expression of *SHOX* was decreased when WRN was mutated in the helicase domain (Supplementary Fig. 12a, b). Immunofluorescence staining of SHOX confirmed these findings

(Fig. 9a). Again, *wrn*⁻/⁻ zebrafish were microinjected with the various WRN mutant mRNA, and in line with the in vitro findings, the loss of WRN helicase failed to facilitate the expression of *shox* in vivo (Fig. 9b).

Driven by the above observations, how WRN helicase regulated *SHOX* expression was further examined. WRN-ChIP-seq peak distribution also showed that WRN preferred to bind to the promoter region

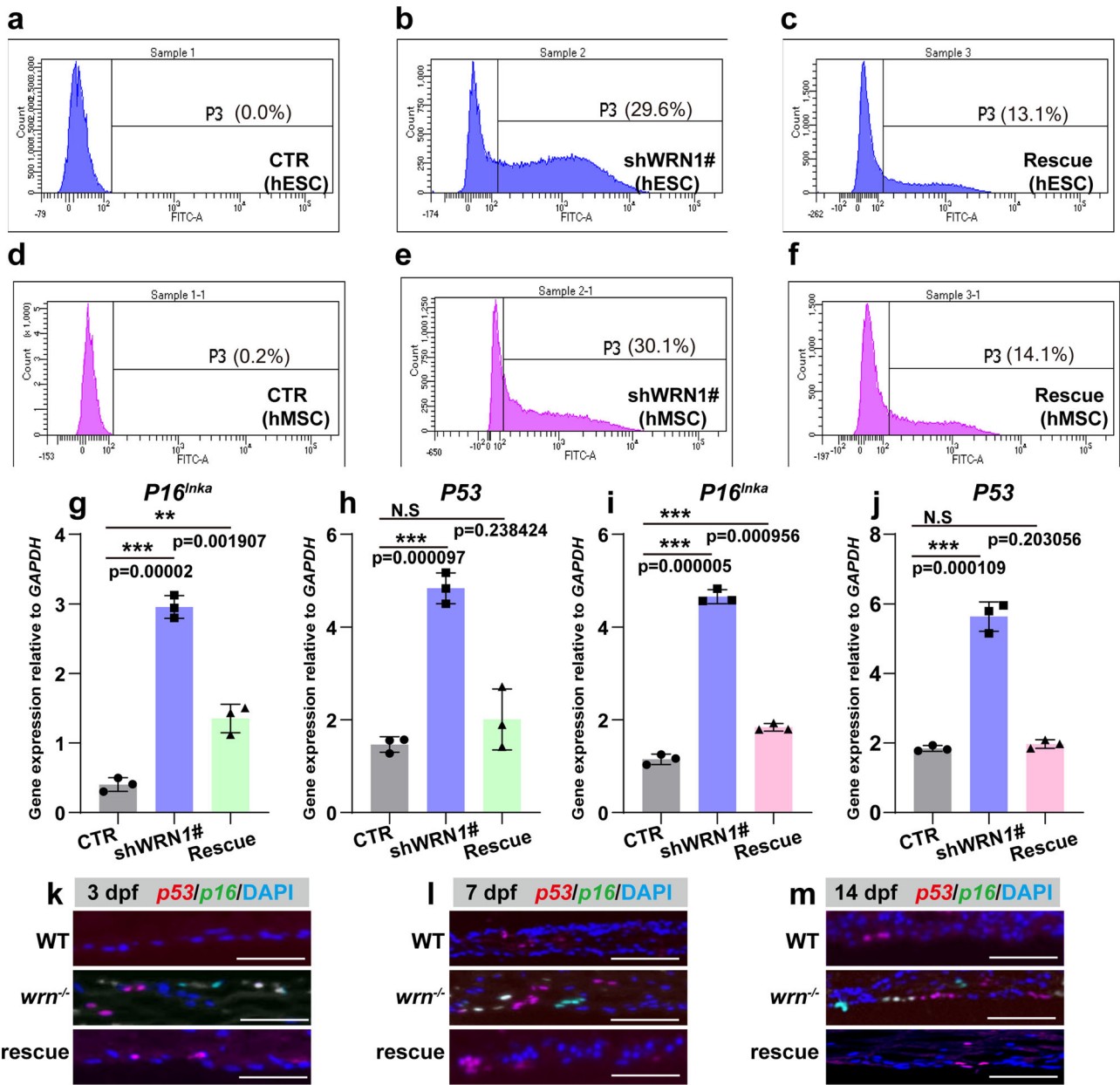

**Fig. 8 | SHOX/shox prevents chondrocyte senescence in WS models. a**–f Representative flow cytometry plots of three independent biological experiments for senescent analysis among CTR, shWRN1#, and rescue groups in hESC (**a–c**) and hMSC (**d–f**). **g–j** qRT-PCR measurement of *P53* and *P16[Ink4a]* in CTR, shWRN1#, and rescue groups in hESC (**g–h**) and hMSC (**i–j**) on day 14. *N* = 3 independent biological

experiments. **k–m** Representative FISH analysis of 3 independent experiments of *p53* and *p16* in WT, *wrn[-/-]* mutant, and rescue groups on 3 dpf, 7 dpf, and 14 dpf. Scale bar = 50 μm. Data are presented as the mean ± S.D. Statistical analysis was performed using two-tailed unpaired Student's *t*-test. **P* < 0.05, ***P* < 0.01, ****P* < 0.001.

(Supplementary Fig. 12c). De novo WRN binding motif analysis showed that WRN preferred to bind to *SHOX* in a guanine-rich region (Supplementary Fig. 12d).

Guanine-rich elements in the human genome are prone to forming G-quadruplex structures (G4s)[43], which are involved in gene transcription and blocking of gene expression[44,45]. Based on previous studies showing the G4 resolving capacity of WRN[46], we speculated that WRN helicase might regulate SHOX expression by unwinding its G4s (Fig. 9c). To confirm it, we firstly tested the existence of G4s using a specific G4 antibody. Through the introduction of different WRN mutant plasmids to WRN-KD cells (shWRN1#), G4's expressions were significantly reduced when full-length WRN or X-WRN was expressed, but not when K-WRN, R-WRN, or F-WRN were expressed (Fig. 9d). This finding suggests that WRN helicase can unwind G4s. As shown by *WRN*-

ChIP-qPCR (Fig. 9e and Supplementary Fig. 13e), WRN could bind to *SHOX*-G4s in the promoter region.

Encouraged by the mechanism of how WRN helicases can open SHOX-G4s, the existence of SHOX-G4s was validated by ChIP-qPCR (Fig. 9f and Supplementary Fig. 13f). The level of G4s was significantly increased in WRN-KD hESCs and WRN-KD hMSCs (shWRN1# cell lines were used) as compared to the CTR cells on day 14, indicating that WRN was important for unwinding G4s. (The IgG was selected as the negative control and the human *GAPDH* primers binding RNA polymerase II were selected as the positive control. Results were shown in Supplementary Fig. 13a–d).

To examine the regulatory effects of *WRN* on SHOX transcription, an oligonucleotide containing human G4 region (h-SHOX-G4) was synthesized with the assistance of QGRS software. We tested the

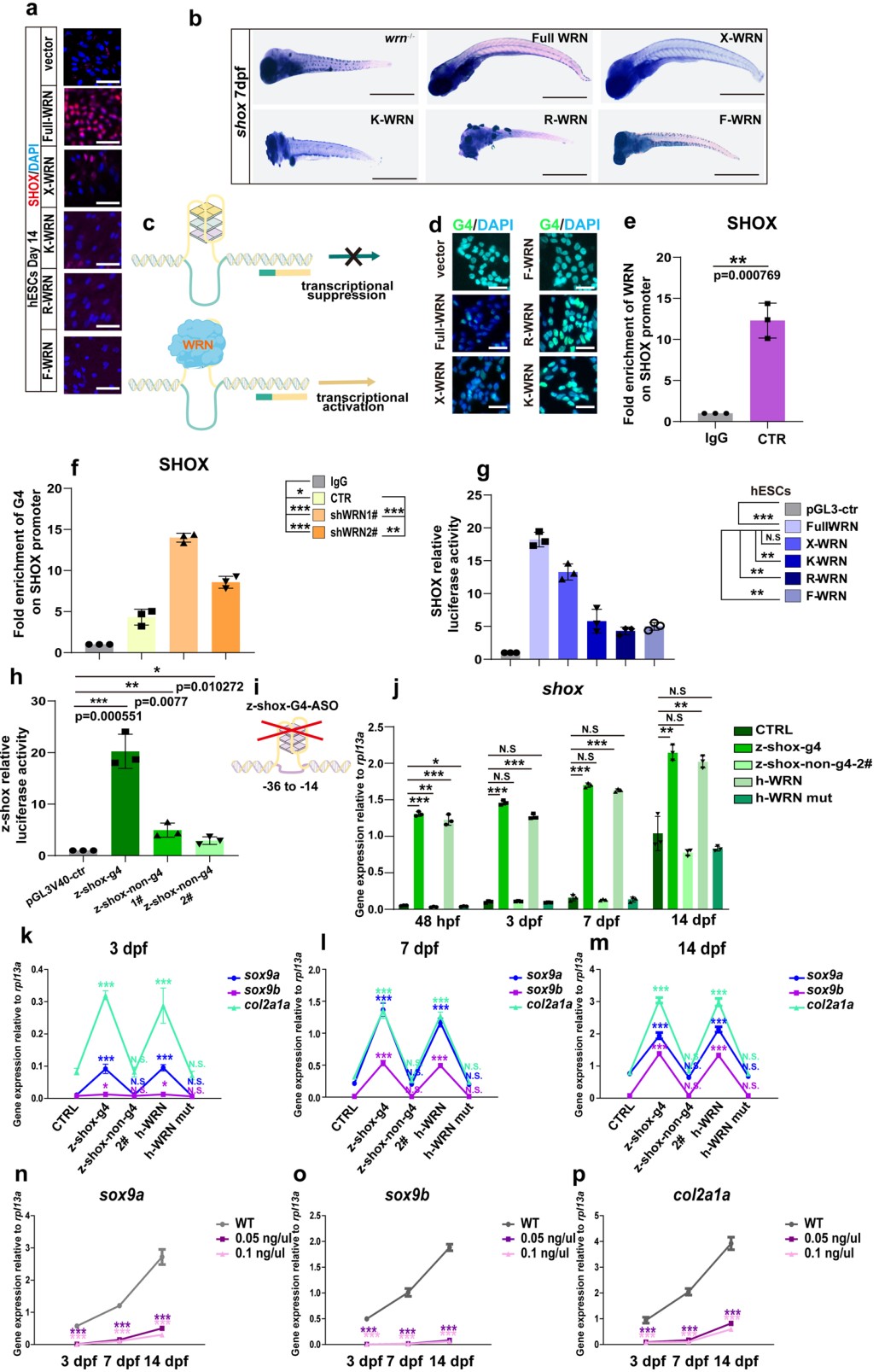

formation of h-SHOX-G4 in vitro in a ThT assay. The fluorescent intensity (F/F$_0$) represented the presence of the G4s (a value of more than 20 is typically regarded to indicate G4 formation). Human c-Myc G4 pu18 was used as the positive control, and the pu18 mutant was used as the negative control according to previous report[47]. The results confirmed the existence of h-SHOX-G4s (Supplementary Fig. 13g) in vitro, which were cloned into pGL3-basic vector. The dual-luciferase activity assay was performed by transfecting h-SHOX-G4 and different WRN plasmids in shWRN1# hESCs and shWRN1# hMSCs, respectively (Fig. 9g and Supplementary Fig. 13h). Interestingly, in comparison with full-WRN, the expression of SHOX in K-WRN, R-WRN, and F-WRN were all significantly reduced. Additionally, an oligonucleotide containing the zebrafish G4 region (z-shox-g4) as well as two oligonucleotides without z-shox-g4 were synthesized, and their

**Fig. 9 | WRN helicase unwinds SHOX G-quadruplexes. a** Representative immunofluorescent staining of 3 independent experiments with different *WRN* mutant plasmids in hESCs on day 14. SHOX was examined. Scale bar = 50 μm. **b** Representative WISH analysis of 3 independent experiments of the expression of *shox* on 14 dpf. Scale bar = 50 μm. **c** Illustration of the mechanism of WRN in opening G-quadruplex. **d** Representative immunofluorescent images of 3 independent experiments with different *WRN* mutant plasmids in hESCs on day 14. G4 was examined. Scale bar = 50 μm. **e** Fold enrichment of WRN on SHOX promoter using ChIP-qPCR analysis. *N* = 3 independent biological experiments. **f** Fold enrichment of G4 on SHOX promoter using ChIP-qPCR analysis. *N* = 3 independent biological experiments. **g** Luciferase assay of the human *SHOX* transcription activity. **h** Luciferase assay of the zebrafish *shox* transcription activity. **i** Illustration of zebrafish *shox-G4* anti-sense oligonucleotides design. **j** qRT-PCR measurement of *shox*. *N* = 3 independent biological experiments. **k**–**m** qRT-PCR measurement of *sox9a*, *sox9b*, and *col2a1a* at 3 dpf (**k**), 7 dpf (**l**), and 14 dpf (**m**). *N* = 3 independent biological experiments. **n**–**p** qRT-PCR measurement of *sox9a* (**n**), *sox9b* (**o**), and *col2a1a* (**p**) at different time points. *N* = 3 independent biological experiments. Data are presented as the mean ± S.D. Statistical analysis was performed using two-tailed unpaired Student's *t*-test. *$P < 0.05$, **$P < 0.01$, ***$P < 0.001$.

formations were confirmed (Supplementary Fig. 14a). They were cloned into pGL3SV40 vector and a dual-luciferase activity assay was conducted by transfecting z-shox-g4 or z-shox-non-g4 with full-length *wrn* plasmids together in shWRN1#−293T cells (the knockdown efficiency was confirmed as Supplementary Fig. 14b shown). It was found that full-length *wrn* facilitated *shox* expression more significantly in the z-shox-g4 group compared with that in the z-shox-non-g4 groups (Fig. 9h), indicating that G4s are crucial in gene transcription in vitro.

However, the actual effect of these G4s in a cellular and genomic natural environment may be different. An important question is whether G4s are truly involved in transcriptional regulation in vivo during development. To answer this question, the G4 ligands were disrupted by microinjecting z-shox anti-sense oligonucleotides (ASO) complementary to the selected G4s into *wrn*−/− zebrafish (Fig. 9i, j). 10 pg-ASO/embryos or controls (an oligonucleotide that did not anneal to the zebrafish genome was used as the control (CTRL)) was injected at the one-cell stage according to a previous report[48]. After confirming the efficiency of ASO (Supplementary Fig. 14c), qRT-PCR was performed and the increment of *shox* expression along with development was observed. In the meantime, it was found that overexpression of h-full-length-WRN mRNAs promoted *shox* expression subsequent to microinjection, whereas h-WRN helicase mutant mRNAs did not. Additionally, it was noted that the disruption of z-shox-g4 enhanced chondrogenesis and similar effect was observed when microinjecting h-full-length-WRN mRNAs (Fig. 9k−m).

In addition, we treated the *wrn*−/− mutant zebrafish with the G4 stabilizer followed by previous reports[49]. The optimal dose of the G4 stabilizer that would not cause death or developmental malformations was firstly determined. One-cell stage *wrn*−/− zebrafish embryos were injected with several concentrations of the G4 stabilizer ranging from 0 ng/μl to 20 ng/μl. Embryos that were dead, deformed, and normal were evaluated at 24 hpf and 48 hpf. As Supplementary Fig. 14d, e shown, 0.05 ng/μl and 0.1 ng/μl were selected for subsequent experiments. The mRNA expression of *shox* was evaluated at different timepoints (3 dpf, 7 dpf, and 14 dpf), and we noted that the mRNA expression of *shox* of *wrn*−/− zebrafish decreased significantly compared to that of the wildtype zebrafish (Supplementary Fig. 14f). Additionally, the expression of specific chondrocyte-associated markers (*sox9a*, *sox9b*, and *col2a1a*) was examined, and we showed that the mRNA expression of these genes was downregulated when treating with the G4 stabilizer in *wrn*−/− zebrafish (Fig. 9n−p). Taken together, these data indicate that WRN regulates *SHOX* transcription and expression through unwinding SHOX-G4s via its helicase activity and promotes chondrogenesis.

## Discussion

Here we report the characterization of *wrn* zebrafish mutants, highlighting the complex mechanism through which WRN is regulated to control chondrocyte development. Our findings have several implications. First, *wrn* mutations lead to short body length and impairment of chondrogenesis in the zebrafish and in stem cell models, providing new models for the studies of WS, especially for studying short stature, which is not obvious in WS mice. Second, WRN plays an essential role

in chondrogenesis primarily through its helicase activity. Third, SHOX functions as a direct downstream player in WRN-regulated chondrogenesis. Finally, WRN helicase binds to the promoter region of *SHOX* and upregulates its transcription by unwinding G4s in the *SHOX* promoter. Thus, our study has elucidated a highly precise and complex mechanism for the short stature seen in WS.

Short stature is a typical clinical sign of WS. Here, we firstly reported that the *wrn* gene plays an important role in early embryonic growth and bone production in zebrafish. Zebrafish is a favored model to study bone development due to the evolutionary conservation of the *wrn* gene and the ease of analyzing early development stages[26]. In our study, we examined the expression of *wrn* at the early development stage and showed that it was expressed in somites, the notochord, and the vertebral column, suggesting that *wrn* was involved in bone development. More importantly, blockade of *wrn* severely delayed embryonic growth and resulted in short stature. The *wrn*−/− mutant zebrafish displayed stunted bone formation in the fish as early as 7 dpf, which might be caused by impairments in the elongation of bones. Longitudinal bone elongation is the engine of human stature growth[34], which is a coordinated and orchestrated process including three main types of chondrocytes: resting, proliferative, and hypertrophic chondrocytes[50,51]. Previously, *WRN* was shown to be crucial for cell growth; most of the Hela cells were arrested in S-phase when the WRN protein was dysfunctional[52]. Additionally, cultured WS cells exhibited a limited ability to proliferate and a prolonged S-phase[53]. In this study, FISH assay for chondrogenic markers showed weak signals in *wrn*−/− zebrafish, and BrdU staining and in situ apoptosis results showed more apoptotic signals and fewer proliferative signals in *wrn*−/− zebrafish, suggesting that depletion of *wrn* inhibited bone growth but promoted apoptosis, excluding the possibility that these chondrocytes were negatively selected by apoptosis. It has been reported that apoptosis-associated genes are dysfunctional in mice lacking *Wrn* protein[54]. In the meantime, we found more γH2AX signals appeared in *wrn* zebrafish at early developmental stage, indicating the loss of *wrn* caused DNA damage, which could be one of the plausible explanations that why loss of *wrn* caused apoptosis[55]. Microinjection of human *WRN* mRNA into the one-cell stage of *wrn*−/− mutant embryos could facilitate chondrocyte proliferation and increase total body length, indicating that *WRN* is crucial for chondrocyte growth development.

To further investigate how *WRN* regulates bone development, we used two stem cell models – hESCs and hMSCs – to study cartilage development. For hESCs differentiation method, a chemically defined, efficient, and repeatable protocol was followed[35]. Pluripotent hESCs (stage 1, day 0) were directed towards a primitive streak-mesendoderm population (stage 2, day 4), after which differentiation proceeded to a mesoderm population (stage 3, day 9), and finally towards chondrocytes (stage 4, day 14). The advantage of this hESC model is that it is useful for examining the complete process of chondrocyte differentiation in the context of WS. hMSCs were also used because they were easily encouraged to differentiate into an efficient model of chondrogenesis, and the commitment of mesenchymal cells to the chondrogenic lineage is important for the formation of chondrocytes. The chondrogenic process in hMSC model was highly similar to the hESC model from stage 3 to stage 4.

Therefore, hMSCs could help researchers focus more on chondrocyte-specific growth stages.

We determined the specific stage when *WRN* functions on chondrogenesis. Our findings showed that *WRN* was expressed during the early stage of pluripotency and was specifically up-regulated during chondrocyte differentiation. Ablation of *WRN* did not affect the pluripotential status, which has been corroborated by the previous studies[37]. However, upon differentiation into multipotent MSCs, the differentiating ability was impaired by *WRN* loss somehow. Our data demonstrated that loss of *WRN* inhibited MSC differentiation and chondrogenesis, which agreed with the results obtained in hMSCs in this study, as well as others' findings[56]. In the meantime, to confirm the specific role of WRN in chondrogenesis, we evaluated another helicase protein, BLM and observed no significant difference between the wildtype and *WRN/wrn*-KD groups both in vitro and in vivo. Ken et al. suggested that the functional deletion of either WRN or BLM cannot be compensated by the other proteins (or other RecQ proteins), indicating that WRN and BLM play different roles in cells[57]. These results together suggested that WRN played a pivotal role in the modulation of chondrogenesis.

Helicases play a crucial role in genome maintenance by unwinding structured nucleic acids structures and various malignancies and genetic illnesses linked to helicase defects propelled them to prominence[17,58]. WRN is a DNA helicase in the RECQ family, and RECQ helicase mutants display dysfunction in DNA replication, recombination, and repair, thus indicating a role for RECQ helicases in protecting genomic integrity[21,22,59]. *WRN* helicase uses the energy from ATP hydrolysis to unwind nucleic acid structures, such as G4s and triplexes in the genome[21]. Whether the WRN DNA helicase activity and its unique exonuclease activity coordinate to contribute to accelerated ageing has been a puzzle in the field. Our current study provided a deeper understanding of chondrogenesis in WS by showing that WRN helicase can facilitate cartilage development. Our data showed that the dysfunction of the helicase core activity (ATPase domain or RECQ domain) of WRN severely impaired chondrogenesis; however, the dysfunction of the exonuclease activity slightly affect chondrogenesis, suggesting the importance of WRN helicase during chondrogenesis. Interestingly, using both a *C. elegans* and *wrn-1* (*gk99*) mutant, which harbors a helicase mutation, and a *Drosophila melanogaster* WS model in which the *Wrn* exonuclease domain (*Wrnexo*[RNAi]) is inhibited, it was reported that both the helicase and the exonuclease domains are essential for normal lifespan and healthspan[2,60]. However, the potential role of WRN in chondrogenesis was not explored due to the limitations of these models when working on bone development and growth. In this study, using a combination of both stem cell and zebrafish models, our data clearly show that it is likely that it is the helicase, rather than the exonuclease domain, is pivotal in chondrogenesis and increase in height.

Integrative analyzes of RNA-seq and ChIP-seq data have shed light on direct transcriptional targets of *WRN* in bone homeostasis. The *SHOX* gene, which is located on both X and Y chromosomes in the pseudoautosomal region 1 (PAR1), consisting of 6 exons, was identified[61]. The *SHOX* gene is widely found in vertebrate species, including chimpanzees, chickens, and zebrafish, with the notable exception of rodents, which have lost the gene over the course of evolution[61]. Several studies have shown that SHOX plays a fundamental role in determining human height, and SHOX deficiency is the most common genetic developmental defect related to isolated and syndromic types of short stature[62,63]. However, no studies have reported a role for *SHOX* in chondrogenesis from the pluripotential stages, and the regulatory mechanism is unclear. Our results showed that the pluripotency of cells was not influenced by the depletion of *SHOX* or *WRN* during chondrocyte differentiation from hESCs, and the expression pattern of *SHOX* was similar to that of *WRN*, pinpointing that *WRN* and *SHOX* synchronously regulated chondrogenesis in WS. In our study, the ablation of *shox* led to bone formation defects and short

stature both in vivo and in vitro, which is in agreement with other studies showing that morpholino-mediated *shox* silencing resulted in significant growth retardation (decreased somite numbers and shortened body length) as well as impaired ossification in the anterior vertebrae[64]. Notably, both in vitro and in vivo loss of function assays showed that the depletion of *WRN* led to the loss of S*HOX*. Our data showed that *SHOX* and *WRN* had a highly similar expression pattern during chondrogenesis, and overexpression of *SHOX* could rescue the chondrogenesis deficiency induced by *WRN*-KD both in vitro and in vivo. More intriguingly, we found that overexpression of *SHOX* in *wrn*[−/−] zebrafish could enhance the expression of other sox family as well, such as *sox5* and *sox6*. SHOX, SOX5, and SOX6 were co-expressed in 18-and 32-week human fetal growth plates, indicating that SHOX could coregulate these genes in skeletal development and body stature[65]. Additionally, we noted that loss of WRN/wrn led to the accumulation of senescent cells both in the zebrafish and in vitro stem cell culture model systems. And overexpression of *SHOX* mitigated the senescent phenotype both in vitro and in vivo. Taken together, these results suggested that *SHOX* could be the direct downstream effector of *WRN* in modulating chondrogenesis.

Noncanonical secondary structures produced in guanine-rich DNA and RNA sequences are known as G-quadruplexes (G4s)[66]. Recent progress in G4s studies has shown that the G4s are likely to be involved in modulating numerous biological processes and interactions with certain protein factors, such as helicases[45]. A computational study indicated the enrichment of putative quadruplex sequences (PQSs) in promoters, especially proximal to the transcription start sites (TSSs), and around 43% of human gene promoters contain one or more G4s[67]. Our global WRN ChIP-seq binding motif results showed that the chromatin binding sites of *WRN* were guanine rich, potentially predisposing them to a high possibility for formation of G4s, which could be regarded as robust proof that *WRN* could bind to G4s. Our SHOX-G4 ChIP-qPCR findings showed the presence of G4s in the *SHOX* promoter region. More excitingly, the luciferase assay showed that the WRN helicase could modulate *SHOX* transcriptional activity. It is reported that WRN has a preference to bind with a "bubble" structure, such as duplex DNA and G4 quadruplex, and functions as a helicase to open these structures to promote DNA transcription, duplication, and replication. However, does WRN really has a preference on resolving G4 structures? Our results showed that *wrn* could promote *shox* expression significantly, especially when the group with a G4 structure was compared with non-G4 structure groups, suggesting that the WRN helicase was sensitive to recognize a G4 structure. Collectively, our results reveal a previously unknown function of WRN in regulating chondrogenesis through transcriptional regulation of SHOX via the WRN helicase domain, highlighting the possibility for the development of therapeutic strategies to treat both this incurable disease and possibly other diseases displaying short stature.

## Methods

### Cell culture and in vitro cell differentiation

hESCs (H1) were maintained on Matrigel-coated plates (Corning) in mTeSR medium (STEMCELL) prior to differentiation. At 70–80% confluency, the cells were passaged at a ratio of 1:3 to 1:4 using Dispase (STEMCELL) enzyme.

hMSCs were cultured with Dulbecco's modified Eagle's medium (DMEM, GIBCO, 1 g/L D-glucose) with 10% MSC-FBS (Thermo Fisher, Cat. 12662029), 1% PSA (GIBCO), and 1% L-glutamine (100x, GIBCO). At 70–80% confluency, cells were passaged at a ratio of 1:3 to 1:4 using 0.25% Trypsin-EDTA (GIBCO).

Chondrocyte differentiation from hESCs was performed following a previously published protocol (35). Briefly, after 2 days of culture in mTeSR when the cells reached 80–90% confluency, mTeSR was replaced with basal differentiation medium including DMEM/F12, 2 mM L-glutamine, 1% (vol/vol) ITS (Life technologies, 41400), 1% (vol/vol)

nonessential amino acids (Life technologies, 11140), 2% (vol/vol) B27 (Life technologies, 17504), and 90 μM β-mercaptoethanol (Life technologies, 31350) supplemented with key growth factors and chemicals as appropriate including WNT3A, NT4 (R&D systems), BMP4, GDF5, Activin-A (Peprotech), FGF2 (Biosource, Invitrogen), and Follistatin 300 (Sigma).

Chondrocyte differentiation from hMSCs was followed using the StemPro Chondrogenic Differentiation Kit (GIBCO, Invitrogen, Grand Island, NY). Briefly, hMSCs were generated from micromass cultures by seeding a 5 μl droplet of cell solution ($1.5–2 \times 10^7$ cells) in the center of the plates, and the cells were allowed to attach in the incubator for 2 h before gently adding chondrogenesis medium. The medium was changed every 2–3 days.

## Zebrafish Maintenance
All zebrafish husbandry and maintenance procedures were approved by the Chinese University of Hong Kong Animal Ethics Committee (AEEC, Ref No. 20-200-MIS). The F1 mutants, *wrn*[sa34829] (C > T) and *shox*[sa41471] (A > T), using ENU method, were purchased from The Zebrafish Information Network (ZIRC, zirc.org).

## RNA-seq and quantitative RT-PCR (qRT-PCR)
Total RNA was extracted using TRIzol™ Reagent (Invitrogen) following the manufacturer's instructions. Reverse transcription was performed using a MasterMix kit (Takara) following the manufacturer's instructions. Quantitative polymerase chain reaction (qRT-PCR) was performed using a Universal SYBR Green MasterMix (Takara) on a QuantStudio 7 Flex real-time PCR system (Applied Biosystems). Gene expression was normalized to human *GAPDH* or zebrafish *rpl13a*. Primers were shown in supplementary table 1. For RNA-seq, whole transcriptome expression was measured using the NGS platform from Novogene according to Novogene's suggested procedures for library construction and data analysis. Bioinformatic analysis was provided by Novogene (Beijing, China).

## Full-length WRN/wrn plasmids and WRN mutagenesis plasmids
Human *WRN* cDNA was synthesized and cloned into the pCDH-EF1 vector (SBI, CD550A-1) between EcoRI and NotI restriction sites. WRN mutagenesis followed the Q5 site-directed mutagenesis instructions (NEB, E0554) Different WRN mutagenesis PCR primers were designed using the NEB online design software NEBase Changer™ (http://nebasechanger.neb.com/) and can be found in Supplementary table 1.

Zebrafish *wrn* cDNA was synthesized and cloned into the Z-pBluescript II vector (provided by Prof. Zhao Hui, School of Biomedical Sciences, the Chinese University of Hong Kong, Hong Kong) between EcoRI and BamHI restriction sites. Primers were synthesized in BGI Genomics (Beijing, China) and can be found in Supplementary table 1.

## Viral preparation and transduction
To knock down WRN and SHOX, short-hairpin RNA (shRNA) sequences were cloned into pLKO.1-TRC cloning vector (Addgene, plasmis#10878) under the control of the U6 promoter. CTR (shNC, non-target any known mammalian genes) and shRNA oligo sequences were synthesized in BGI Genomics (Beijing, China) and can be found in Supplementary Table 1.

Lentiviruses were prepared using 293 T cells as described[68]. Briefly, HEK293T cells were seeded at $5 \times 10^6$ cells per 10 cm dish and incubated for one day. shRNA expression vector and the lentiviral packaging constructs pMD.2 G (Addgene, plasmis#12259) and psPAX2 (Addgene, plasmis#12260) were con-transfected into HEK293T cells with 1 mg/ml PEI solution (Polysciences, Cat#23966-1). Lentiviral supernatants were collected at 48 h and 72 h after transfection. Lentiviruses were harvested by ultracentrifugation for 3 h at 60, 000 *g* with a himac CR/22 G centrifuge at 4 °C.

## Alcian blue staining
The culture medium was discarded, and the cells were washed with PBS three times, and then immersed with Alcian blue (Sigma, B8438) for 20 min. After immersion, cells were washed with PBS three times to remove the excess dyes. Cells were further for image analysis.

## Masson's trichrome staining
Masson's trichrome staining was performed as described (http://www.ihcworld.com/_protocols/special_stains/masson_trichrome.htm). Briefly, the sections were deparaffinized and rehydrated through 100% ethanol, 95% ethanol, 70% ethanol, and then washed under running tap water for 1–5 min. The sections were stained with Weigert's iron haematoxylin working solution for 10 min and washed in running tap water for 1–5 min. Then the sections were stained with Biebrich scarlet-acid fuchsin solution for 10 min and washed in running tap water for 1–5 min. The sections were differentiated in a phosphomolybdic-phosphotungstic acid solution for 10–15 min. The sections were directly transferred to aniline blue solution without rinse for 5–10 min, then washed in running tap water for 1–5 min, and differentiated in 1% acetic acid solution for 2–5 min. Dehydration of the sections continued through 95% alcohol, 100% alcohol, and cleaned xylene. Finally, the sections were mounted and imaged with a Nikon Ni-U upright microscope.

## Immunofluorescence staining
Cells at different stages of differentiation were subjected to immunostaining by first being fixed with 4% formaldehyde for 20 min at room temperature (RT). Cells were washed with PBS and permeabilized with 1% Triton X-100 / PBS for 30 min at RT. After blocking with 1% bovine serum albumin at RT for 1 h, cells were incubated with primary antibodies [anti-WRN (1:100 dilution, Sigma, W0393), anti-SHOX (1:100 dilution, Thermo Fisher, PA5-65140), anti-SOX9 (1:100 dilution, R&D systems, AF3075), anti-Collagen II (1:100 dilution, Santa Cruz Biotechnology, sc-7764), anti-Ki-67 (1: 100 dilution, BD), and anti-G4 (1: 200 dilution, Merck, MABE 1126)] in 4 °C overnight. After that, cells were incubated with selected Alexa secondary antibodies for 1 hr at RT. Alexa Fluor 488-conjugated anti-mouse IgG (1: 1000 dilution, Invitrogen, A28175), Alexa Fluor 488-conjugated anti-rabbit IgG (1: 1000 dilution, Invitrogen, A11008), and Alexa Fluor 594-conjugated anti-rabbit IgG (1: 100, dilution, Invitrogen, A11037). Cells were incubated with Hoechst 33342 (Invitrogen, H3570) for 10 min at RT. Coverslips were mounted and slides were imaged with a Leica FV 1200 microscope.

Primary antibody [anti-γH2AX antibody (1: 200 dilution, Gentex, GTX127342)] was used for zebrafish embryos immunofluorescent staining. Alexa Fluor 594-conjugated anti-rabbit IgG (1: 100, dilution, Invitrogen, A11037) was used as the secondary antibodies.

All antibodies used were shown in Supplementary Table 2.

## Chromatin immunoprecipitation (ChIP)-qPCR and ChIP sequencing (ChIP-Seq)
The ChIP assay was performed using a Pierce Magnetic ChIP kit (Thermo, 26157). Briefly, cells were fixed with a final concentration of 1% formaldehyde for 10 min at RT. Glycine solution was then added for 5 min at RT. Formaldehyde/glycine-containing medium was aspirated and the cells were rinsed twice with ice-cold PBS. The PBS was removed and ice-cold PBS with 10 μl of the Halt Cocktail was added. The cells were scrapped and transferred to a new microcentrifuge tube, and centrifuged at 3,000 g for 5 min. 200 μl of Membrane Extraction Buffer containing protease/phosphatase inhibitors was added, and the tube was vortexed for 15 sec and incubated on ice for 10 min. The supernatant was removed after centrifuging at 9,000 g for 3 min. Nuclei were resuspended in 200 μl of MNase Digestion Buffer Working Solution with 2 μl diluted MNase (ChIP grade, 10 U/μl), and the tube was vortexed and incubated in a 37 °C water bath for 15 min. 20 μl of

MNase Stop Solution was added to stop the reaction. The tube was centrifuged at 9,000 *g* for 5 min, and 100 µl of 1×IP Dilution Buffer with protease/phosphatase inhibitors was used to resuspend the nuclei. The nuclear membrane was sonicated on ice with several pulses. Between pulses, the tubes were incubated for 30 sec on ice. The supernatant, which contains the digested chromatin, was transferred to a new tube and centrifuged at 9,000 *g* for 5 min.10 µl of the supernatant containing the digested chromatin was transferred to a new tube as the 10% total input sample from one ChIP. The remaining 90 µl of supernatant was transferred to 410 µl of 1×IP dilution buffer. IP reactions were incubated with primary antibodies (WRN, Sigma, W0393) overnight at 4 °C with mixing. Next day, 20 µl of ChIP grade protein A/G magnetic beads were used to enrich the protein-chromatin according to the manufacturer's instructions. 150 µl of 1×IP elution buffer was added to the washed beads, and the tubes were capped and incubated at 65 °C for 30 min with vigorous shaking. During the elution step, 1.5 mL microcentrifuge tubes containing 6 µl of 5 M NaCl, and 2 µl of 20 mg/mL proteinase K were prepared for each IP reaction. The 10% total input sample(s) was thawed, and 150 µl of 1×IP elution buffer, 6 µl of 5 M NaCl, and 2 µl of 20 mg/mL proteinase K were added. Following the 65 °C incubation, the beads were collected with a magnetic stand. The supernatant containing the eluted protein-chromatin complex was transferred and dispensed into prepared tubes with the NaCl and proteinase K. All IP and total-input samples were incubated at 65 °C for 1.5 hr. 750 µl of DNA binding buffer was added to each eluted IP and total input sample, and each sample was transferred to the DNA clean-up column, centrifuged at 10,000 g for 1 min and the flow-through was discarded. 750 µl of DNA column wash buffer was added and centrifuged at 10, 000 g for 1 min and the flow-through was discarded. The column was washed with wash buffer, and the DNA was eluted with 50 µl elution buffer. The resulting solution is the purified DNA. These samples were later used for further experiments, such as qRT-PCR or ChIP-sequencing.

### G4-ChIP qPCR Primer Design

QGRS mapper (https://bioinformatics.ramapo.edu/QGRS/index.php) and DNA analyzer (bioinformatics.ibp.cz/#/) prediction results were used to design the ChIP-qPCR primers for identifying G4s. Combined with the G4 high–throughput sequence analysis and online prediction, transcription start sites (TSS) from the –318 to −289 region predicted a high score of human G4s that were identified.

### Whole-mount in situ hybridization (WISH)

WISH was carried out as described[69]. Embryos were fixed at the appropriate stages with 4% PFA at 4 °C overnight. The next day, the fixed embryos were washed three times with PBST.

On day 1, the embryos were rehydrated in 75%, 50%, and 25% methanol in PBST on a shaker for 5 min at RT and washed three times with PBST. Afterwards, the embryos were permeabilized with proteinase K at RT for the time indicated for the different stages. The proteinase K digestion was stopped by incubating in 4% PFA for 20 min, and washed three times with PBST. The embryos were immersed in hybridization medium (HM, 50% formamide, 5×SSC, 50 µg/mL heparin sodium salt, 0.1% Tween-20, and 5 mg/mL torula RNA) at 65 °C for 2–4 h. HM was replaced with new HM-containing probes. The embryos were incubated at 65 °C overnight.

On day 2, the HM was gradually changed to 2×SSC through a series of 10 min washes at 65 °C in HM diluted with 2×SSC via single washed in stepwise concentrations: 75% HM, 50% HM, 25% HM and 100% 2×SSC. 0.2×SSC was gradually replaced with PBST through a series of 10 min washes in 0. 2×SSC diluted with 1×PBT using single washes in stepwise concentrations: 75% 0. 2×SSC, 50% 0. 2×SSC, 25% 0. 2×SSC and 1×PBST. The embryos were incubated for 3-4 h at RT in blocking buffer. Diluted anti-DIG antibody was added and the embryos incubated at 4 °C overnight.

On day 3, the antibody solution was discarded and the samples were washed in PBST for six times. The embryos were incubated in alkaline tris buffer. Staining buffer was incubated for the appropriate time, and the reaction was stopped and the samples were imaged using an Olympus SZX16 microscope.

Candidate genes were cloned into the DIG-labeled RNA probe was synthesized with T7 or T3 RNA polymerase (Promega, P207B, or P208C).

Zebrafish probes were *sox9a*, *col2a1a*, *col10a1a*, *col1a1a*, *wrn*, and *shox*.

### Fluorescence in situ hybridization (FISH)

The fluorescent-labeled probes were generated with a fluorescence in situ hybridization kit (Thermo Fisher, Cat No. F32956). Briefly, in vitro transcription (for RNA probes) is used for enzymatic incorporation of amine-modified nucleotides, followed by chemical labelling with amine-reactive Invitrogen™ AlexaFluor™ dyes. For in situ hybridization, the sections were processed as WISH described. The images were processed with a Leica TCS SP8 confocal microscope.

Zebrafish probes were *sox9a*, *sox9b*, *col2a1a*, *col10a1a*, *col1a1a*, *wrn*, *shox*, *p53*, and *p16*.

### Microinjection

The *wrn* mutant embryos were collected in one-cell stage for micro-injection of 300 pg *shox* mRNA or *wrn* mutant mRNA. In vitro transcription with SP6 mMessage mMachine SP6 Kit (Ambion, Cat. No. AM1340).

10 pg-ASO/embryos or CTRL was injected into the one-cell staged zebrafish embryos (Previous report had confirmed this optimal concentration by testing the survival of specimens until 48 hpf[48]).

### In vitro apoptosis assay

Cells were stained following to the cell apoptosis kit (V35117, Invitrogen) instructions. Briefly, prepare the poly-caspases apoptosis working solution by adding 1 part of FLICA reagent stock to 4 parts PBS. Cells were incubated with apoptosis dyes and PI working solution (100 µg/mL) for 15–20 min at room temperature. After the incubation, cells were washed for three times with PBS. Then cells were stained with Hoechst 33342 (Invitrogen, H3570) for 10 min at RT and washed for three times with PBS. The samples were further analyzed with Nikon Eclipse TS2 microscope.

### BrdU Treatment

Larval or juvenile fish were treated with 4.5 mg/mL BrdU (Sigma Aldrich, B5002) by bath application for 1-3 hr depending on the days post fertilization, followed by several times washes. The fish were then euthanized and for further analysis.

### Luciferase reporter assay

Single-stranded oligodeoxyribonucleotides were synthesized from BGI (Beijing, China). Duplex human-G4-DNAs were generated by annealing oligonucleotides representing G4 regions. Then the duplex human-G4-DNAs containing SamI and XhoI restriction sites were cloned into a pGL3-basic vector (provided by Prof. Zhao Hui, School of Biomedical Sciences, the Chinese University of Hong Kong, Hong Kong). WRN-KD hESCs or hMSCs were transfected with 0.5 µg human-SHOX-G4 pGL3 luciferase reporter constructs plasmids and different WRN mutant plasmids. Renilla Luciferase plasmids (provided by Prof. Zhao Hui, School of Biomedical Sciences, the Chinese University of Hong Kong, Hong Kong) were co-transfected as an internal control. After 48 hr transfection, cell lysates were collected for luciferase assays with a luciferase substrate system manufacturer's instruction (Promega E1910, America).

Duplex zebrafish-G4-DNAs were generated by annealing oligo-nucleotides representing G4 regions. Then the duplex zebrafish-G4-

DNAs containing SamI and XhoI restriction sites were cloned into a pGL3-SV40 promoter plasmid (Promega, U47298). WRN-KD 293 T cells were transfected with 0.5 µg zebrafish-SHOX-G4 pGL3 luciferase reporter constructs plasmids and full-length zebrafish *wrn* plasmids. Renilla Luciferase plasmids were co-transfected as an internal control. After 48 hr transfection, cell lysates were collected for luciferase assays with a luciferase substrate system manufacturer's instruction (Promega E1910, America).

### Senescence flow cytometry assay
Cellular senescence assay is followed by CellEvent™ senescence green flow cytometry assay kit (C10840) instructions. Briefly, the cells were washed and resuspended in 1× PBS to a concentration of $0.5 \times 10^6$ to $1.0 \times 10^6$ cells per 100 µl. The cell suspension was aliquoted to 100 µl per tube and centrifuged, then the media was discarded. The cells were resuspended in a fixation solution and incubated for 10 mins at room temperature, protected from light. The cells were washed with 1% BSA in PBS and resuspended in a working solution, incubated for 1-2 hr at 37 °C. After incubation, the cells were washed with 1% BSA in PBS. The cells were resuspended in PBS and analyzed on a flow cytometer BD LSR Fortessa cell analyzer using a 488-nm laser. In total 100, 000 cells were analyzed per measurement. Data were analyzed using BD DACSDiva 9.3 software.

### In vivo apoptosis assay
In vivo apoptosis assay was followed Click-iT™ Plus TUNEL Assay (C10618) instructions. Briefly, deparaffinize tissue sections and then the slides were immersed in fixative (4% paraformaldehyde) for 15 min at 37 °C. The slides were washed in PBS for 5 min each. The slides were completely covered with permeabilization reagent and incubated for 15 min. The slides were washed in PBS for 5 min and immersed in the fixative for 5 min at 37 °C. After washing the slides, TdT reaction buffer are added and incubated for 10 min at 37 °C. The TdT reaction buffer were removed and the prepared TdT reaction mixture were added and incubated for 1 hr at 37 °C. The slides were washed in PBS containing 3% BSA and 0.1% Triton™X-100 for 5 min. The Click-iT™ Plus TUNEL reaction cocktail were added and incubated for 30 min at 37 °C, protected from light. The slides were washed with 3% BSA in PBS for 5 min. 1xHoechst 33342 solution (H3570, Invitrogen) was added to the slides and incubated for 15 min at room temperature, protected from the light. The samples were washed in PBS and processed for image analysis a Leica TCS SP8 confocal microscope.

### ThT (Thioflavin T) fluorescence assay
ThT fluorescence assays were conducted followed by previous protocols[47]. Briefly, oligonucleotides were heated at 95 °C for 5 min at 2 µM concentration in 100 mM Tris-HCI (ph = 7.5), 100 mM KCI and cooled down to room temperature. Oligonucleotides and ThT (Sigma T3516) were mixed at 1 µM final concentration in 50 mM Tris-HCI and 50 mM KCI in a volume of 200 µl using 96-well microplates (Genetimes). Fluorescence emission measurements were checked using a microplate reader (SpectraMax M3) with emission spectrum ($\lambda_{ex}$ 485 ± 20 nm, $\lambda_{em}$ 528 ± 20 nm). The fluorescence enhancement was defined as the ratio between ThT fluorescence in the presence of oligonucleotide (F) and background fluorescence of ThT alone ($F_0$) after subtraction of the buffer fluorescence. Pu18 oligonucleotides representing G-quadruplex from human c-Myc promoter was used as the positive control, and a mutated version of pu18 that destroys G-quadruplex formation as the negative control following the previous report[48].

### G4 stabilizer treatment
The G4 stabilizer was purchased from MedChemExpress (HY-15176A). 5 nl of the G4 stabilizer was injected with several concentrations ranging from 0 ng/µl to 20 ng/µl to determine the optimal dose. The embryos at 24 hpf and 48 hpf were collected for further analysis.

### Calcein green staining
Calcein green staining was performed following a previous protocol[70]. In brief, 2 g of calcein powder (Sigma,Cat#C0875) was dissolved in 1 mL of deionized water. Zebrafish embryos were netted and immersed in the solution for 3 to 15 mins, depending on the days post fertilization. After staining, the embryos were rinsed several times to remove excess and unbound solution completely. The embryos were then euthanized and imaged with SZX 16 microscope.

### Data collection and statistical analysis
Two-tailed unpaired Student's *t*-test or One-Way ANOVA were used for multiple comparisons. All data were presented as means ± S.D as indicated with a *$P < 0.05$ considered statistically significant. GraphPad Prism (version 8.0) was used for statistical analysis.

### Reporting summary
Further information on research design is available in the Nature Research Reporting Summary linked to this article.

## Data availability
All the related data supporting the findings are available within this article or in the Supplementary Information file. The data sets generated and/or analyzed in the study are available from the corresponding authors at a reasonable request. RNA-seq and ChIP-seq raw data and normalized mapped reads are available from the Gene Expression Omnibus (GEO) under accession numbers GSE206214. Source data are provided as a source data file with this paper. Source data are provided with this paper.

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

## Acknowledgements

This study was partly supported by VC Discretionary Fund provided to the Hong Kong Branch of Chinese Academy of Science Center for Excellence in Animal Evolution and Genetics, Acc 8601011, State Ministries Special Budget to support MOE Key Laboratory for Regenerative Medicine (CUHK-Jinan University) Project Code:2622009, and Shandong University-Chinese University of Hong Kong Seed Fund for International Research Collaboration provided to W. Y. C. E.F.F. was supported by Helse Sør-Øst (#2017056, #2020001, #2021021), the Research Council of Norway (#262175, #277813), the National Natural Science Foundation of China (#81971317), Akershus University Hospital Strategic Funding (#269901, #261973), the Civitan Norges Forskningsfond for Alzheimers sykdom (#281931), the Czech Republic-Norway KAPPA programme (with Martin Vyhnálek, #TO01000215), and the Rosa sløyfe/Norwegian Cancer Society & Norwegian Breast Cancer Society (#207819). S.L. has received funding from the European Union's Horizon 2020 research and innovation programme under the Marie Skłodowska-Curie grant agreement No 801133. This study was partially supported by a grant from the Research Grants Council of the Hong Kong Special Administrative Region, China (Project No. 14109920). Y.Y.T. was supported by a Hong Kong PhD Fellowship (PF17-10458). We thanked Core Laboratories of school of biomedical sciences (the Chinese University of Hong Kong) for kind technical assistance and support.

## Author contributions

Study concept, literature search, experimentation: W.Y.C., Y.Y.T., S.L., E.F.F., and H.H.C. Experimentation and analysis: Y.Y.T. and X.L. Material support: W.Y.C., H.Z., N.D.D., and W.N.L. Manuscript drafting: W.Y.C., Y.Y.T., W.M.W., E.F.F., and S.L.

## Competing interests

E.F.F. has CRADA arrangement with ChromaDex, and is consultant to Aladdin Healthcare Technologies, Vancouver Dementia Prevention Centre, and Intellectual Labs. All other authors declare no competing interests.
