## [Peer Review File · Nature Communications]

WRN promotes bone development and growth by unwinding SHOX-G-quadruplexes via its helicase activity in Werner SyndromeREVIEWER COMMENTS

Reviewer #1 (Remarks to the Author):

This work by Tian et al., focuses on Werner syndrome (WS), a monogenic autosomal recessive disorder caused by mutations of WRN gene. Patients affected by WS are characterized by short stature alongside to other phenotypes. Using a zebrafish *wrn*^{-/-} mutant as a model for WS herein, the authors unveil that *wrn* is a regulator of bone development and growth through transcriptional regulation of *Shox* gene. Moreover, they provide evidence that *Wrn* protein triggers this function by unwinding *Shox*-g –quadruplex via its helicase domain.

Comments:

1. This zebrafish mutant appears to be a good WS model, and the study focuses exclusively on the skeletal phenotype in order to unveil the underlying molecular mechanism(s). Do the authors have any knowledge if this zebrafish *wrn*^{-/-} mutant features any other phenotypes known to be associated to WS? (e.g. premature replication senescence; increased chromosome instability; increased apoptosis; skin defects, like thinning). These would be valuable information since the mouse homolog mouse mutant does not phenocopy so effectively the WS.
2. Figure 2, Panels o-v showing the extent of cartilage formation and proliferative status between WT and *Wrn*^{-/-} mutant reveal less cartilage differentiation as well less proliferative activity in the mutant. How the authors comment this observation? Proliferation and differentiation are usually two mutually exclusive events during development.
3. Figure 3 and Sup Fig.2, show cartilage formation, under the material and methods section the authors when describe the chondrogenic procedure state that chondrocyte differentiation from hESCs was performed following a previously published protocol (32), referring to Bianco P et al. Skeletal stem cells. Development 142 1023-1027 (2015). This paper is a review and importantly, Bianco and Robey in the context of chondrogenic differentiation assay procedure to follow state that "Cartilage formation in micro-mass or pellet cultures is more reliable, as it rests on ultimate histological proof of genuine cartilage". The authors have performed their chondrogenic assay on cell-monolayer instead....
4. Apoptosis is known to be a crucial step during endochondral ossification, additionally misregulation of genes involved in apoptosis has been reported in mice lacking a functional *Wrn* protein and *Wrn* knockdown inhibits proliferation while promoting apoptosis. However, the authors have completely ignored this aspect. Therefore, investigation on apoptosis would be desirable in order to rule out that potential cell competent to specify towards the chondrogenic lineage are negatively selected by apoptosis.
5. Supp. Figure 4, Panel g, illustrates FISH analysis of *shox* expression in WT zebrafish and *wrn*^{-/-} mutant zebrafish 7 dpf. It would be of interest to analyze the expression at other time points, such as dpf 3 and dpf 14 and in parallel to the expression of *wrn*. This profiling would provide better and complete information regarding the co-expression of these two genes during chondrogenesis.
6. Figure 7, panels k-m and u-w show FISH analysis for *Col2a1a* and *col10a1a* expression combined with BrdU staining. The authors comment *Shox1* overexpression in *wrn*^{-/-} zebrafish "significantly enhanced " the expression of these genes as well the proliferation activity. It is hard to believe that a staining can be quantified as significant.... I would suggest a qRT in order to establish if the outcome is significant. Moreover, in these panels it is hard to appreciate *Sox9* and *Col 10a1a* staining, whereas it is possible for *Col2a1a*.

Reviewer #2 (Remarks to the Author):

Werner syndrome (WS) is a rare form of accelerated ageing disease with clinical characteristics resembling those of normal ageing. WS is caused by multiple mutations in the gene encoding the Werner DNA helicase (WRN), but the underlying mechanisms driving short stature in WS patients remains elusive. In this paper, Tian et al., investigated the role of WRN gene in regulating of chondrogenesis and the underlying mechanisms using zebrafish as an in vivo model. The authors found that WRN deficiency results in the inhibition of bone growth and short stature. Moreover, the authors, using in vitro cultured human embryonic stem cells (hESCs) and human mesenchymal stem cells (hMSCs), demonstrated that WRN deficiency impairs cartilage development. The authors also performed RNA-seq and ChIP-seq and showed that the SHOX (short-stature homeobox) gene is a direct target of WRN. WRN regulated SHOX expression by facilitating gene transcription. In addition, the authors demonstrated that manipulating the expression of SHOX is sufficient to mimic or rescue chondrogenesis and bone formation in WT or WRN knockdown models. The authors should be commended on their well-conducted study and engaging manuscript, which contains an impressive amount of data that are interpreted to support some interesting conclusions. I have the following points for the authors to address:

Major comments

It has been recognized that DNA damage, cellular senescence, telomere attrition, and impaired autophagy/mitophagy resulting from mutations in WRN explain the majority of the clinical features of WS. It is somewhat disappointing that these hallmark changes were not evaluated in the testing model systems. Particularly, from the BrdU labeling (Figure 2) and the RNA-seq data (Figure 5), cell proliferation/cell growth-associate genes are among the most significantly downregulated pathways in the WRN deficiency cells (vs. WT cells), suggesting that cellular senescence is likely involved in WRN deficiency-caused bone development and growth defects. A previous report, which is closely related to the current study, showed that stem/progenitor cells in metaphysis of long bone during rapid growth period are highly proliferative but undergo progressive cell senescence during late puberty when bone growth slows down (Nat. Commun. 2017; 8:1312), indicating that cellular senescence is a key mechanism to control bone growth. Therefore, it would be interesting for the authors to assess whether loss of WRN causes accumulation of senescent cells in the zebrafish and the in vitro stem cells culture model systems. It is also interesting to test whether overexpressing SHOX prevented the senescence phenotype.

Minor points

1. The authors should give an overview in the Introduction section on the current understanding of the cellular and molecular mechanisms that result in the accelerated aging of WS patients.
2. The global WRN deficiency models was used in the study. To better demonstrate WRN primarily functions in early-stage chondrocyte/MSCs, it would be helpful to conduct double-immunostaining of WRN/SHOX with chondrocyte/MSC markers in the model system.
2. In Figure 5d, f, it seems SHOX is not the most down-regulated gene according to the z-score and the heatmap. However, SHOX is the most significantly downregulated genes in Figure 5g, h. Can the authors explain the discrepancy?
3. In the ChIP-qRT-PCR data (Figure 8g, h), it would be better if the authors show the PCR gel images with negative and positive controls.

Reviewer #3 (Remarks to the Author):

Summary:

Skeletal dysplasia and short stature are clinical phenotypes of considerable complexity with the large number of genes and spectrum of clinical disorders. SHOX is a key regulator of chondrocyte biology, as reflected by the multiple number of target genes that it regulates. While zebrafish serve as a useful model for whole body characterization of genes regulating height and skeletal components, the pathways are quite complex. Moreover, relating skeletal phenotypes of zebrafish to human and tissue-specific gene regulation is formidable. In the current work, the authors have investigated the hypothesis that the WRN helicase-nuclease implicated in the premature aging Werner syndrome regulates expression of SHOX via its G4 resolvase activity on SHOX-promoter associated G-quadruplexes.

The authors document shortened body length and bone abnormalities in *wrn*^{-/-} zebrafish. From these studies they transition to human embryonic stem cells and human mesenchymal stem cells to examine the role of WRN in chondrogenesis at the cellular level. However, the direct connection of their observations remain elusive to the zebrafish model findings. To their credit, the authors assess if human WRN mRNAs injected into *wrn*^{-/-} mutant zebrafish at the one-cell stage on regulation of expression of key genes in the chondrogenesis pathway. By this analysis, they find that SHOX expression is regulated by WRN during chondrocyte differentiation, and this is largely dependent on WRN helicase activity. SHOX expression on stature was further examined by microinjection of human SHOX mRNA into zebrafish embryos at the one-cell stage and embryo length was correlated with bone growth/development expression. Finally, the authors determine that WRN unwinds promoter-associated G4, thereby regulating its expression.

Critical Comments:

SHOX signaling is notably complex, and there is still only limited understanding of the pleiotropic effects of SHOX deficiency in humans. Therefore, the utilization of model genetic organisms with conserved genes and translational value is appreciated. Zebrafish is one such model for studying chondrogenesis, but has its limitations that are apparent at the organismal and tissue levels. A useful tool in the current study might have been G-quadruplex ligands to modulate SHOX promoter function and interrogate WRN involvement. This experimental approach might have been applied in the *in vivo* model.

Specificity of WRN's effect was not addressed. One wonders if the sequence-related BLM helicase which has also been reported to regulate gene expression via its action on G4 promoters would affect the observed phenotypes via SHOX.

In my mind, it remains unclear if the effect of WRN is specific to regulation of SHOX or if other gene regulatory factors are implicated in genetic determination of height and bone homeostasis are involved, that are also directly regulated by WRN or indirectly. While the study suggests WRN is involved, a direct causal relational and mechanism is not established.

Assessment of WRN connected SHOX gene regulation on other tissues of zebrafish could be performed to provide a better impression of the broadness of the gene regulatory network.

As shown by the authors, 380 genes were found to be regulated by WRN in chondrocyte homeostasis, 116 up-regulated and 264 down-regulated. Given the large number, it may be that multiple genes co-regulated with SHOX or even directly regulated by SHOX are also implicated, as suggested. It would have been informative and supportive of their model if the authors had examined in greater detail the mechanism whereby some of the top candidate involved in chondrogenesis, as evidenced by previous studies, are affected and if their regulation is relevant to the phenotypes of *wrn*^{-/-} zebrafish model in this work. This would apply to SOX9, SOX5, SOX6 (The transcription factors SOX9 and SOX5/SOX6 cooperate genome-wide through super-enhancers to drive chondrogenesis - PubMed (nih.gov)). The gene regulatory networks of chondrocytes are quite complex, and super enhancers as well as histone modifications and chromatin accessibility come into play. It has been proposed that multiple transcription factors are involved in chondrocyte biology. While the evidence from the current study suggests that WRN regulation of SHOX is among these, it is difficult to comprehend the molecular-genetic mode of action in the context of the multiplicity of transcription factors coming into play.

Based on their results, the authors suggest that WRN controls bone growth and development via its regulation of SHOX. Are there other phenotypes beyond bone-related affected by WRN status and are these also mediated by SHOX? The complexity of WRN's genetic linkage to range of clinical features of accelerated aging. This is compounded by the extensive data that at least in vitro (biochemical), cell-based models, and genetic models, WRN plays roles in transcriptional regulation, DNA repair, replication stress response, genomic stability. So is this an oversimplification that WRN's involvement in controlling bone metabolism in zebrafish is mediated by its effect on SHOX, and there is no consideration of its pan-wide functions in multiple cell types and tissues?

Another layer of potential complexity is added with the idea that WRN's key molecular activity of G4 resolution in the promoter of SHOX underlies the growth/bone phenotypes may be overly simplified. WRN, along with other G4-resolving helicases (e.g., BLM) is thought to play a role in regulation of expression of many genes with G4-laden promoters, not to mention its effect on G4 in other chromosomal regions (e.g., telomeres). I'm not sure all the phenotypes related to chondrocyte biology hinge on WRN-catalyzed G4 resolution of SHOX promoter DNA sequence elements. This seems to be the driving conclusion of the paper, and I am not sure that other avenues have been explored which potentially contribute to the observations made for *wrn*^{-/-} zebrafish.

A recently published study by Shin et al., DNA Repair (2021) Large-scale generation and phenotypic characterization of zebrafish CRISPR mutants of DNA repair genes; Large-scale generation and phenotypic characterization of zebrafish CRISPR mutants of DNA repair genes - PubMed (nih.gov) reported that *wrn*^{-/-} zebrafish. Like *mre11*^{-/-} zebrafish and certain other DNA repair mutants displayed reduced growth and development during the juvenile stage. Interestingly, both *mre11acu56/cu56* and *wrnacu64/cu64* failed to survive to 60 dpf. The similar phenotype raises the question if the genetic defect is attributed to unusual G4 metabolism or more generically a DNA repair defect such as double-strand break repair in which both WRN and MRE11 are implicated. It would have been of interest if the authors of the current study had determined what phenotypes and at what developmental stage were observed in *wrn*^{-/-} zebrafish that had been exposed to a G4-binding ligand. If the temporal appearance or severity of the observed phenotypes were G4-inducible, then this would support their model.

Yokokura et al. (Front Endocrinology (Lausanne) 2017; The Short-Stature Homeobox-Containing Gene (shox/ SHOX) Is Required for the Regulation of Cell Proliferation and Bone Differentiation in Zebrafish Embryo and Human Mesenchymal Stem Cells - PubMed (nih.gov)) had assessed the phenotypes of morpholino oligo-mediated knockdown of zebrafish *shox*. Determination if *wrn* and *shox* genetically operate in the same pathway as predicted by the model proposed by the authors of the current study would have helped to strengthen the study.

A 2007 paper by Xie et al in Genetics entitled Manipulating Mitotic Recombination in the Zebrafish through RecQ Helicases Manipulating Mitotic Recombination in the Zebrafish Embryo Through RecQ Helicases (nih.gov) might be relevant to the current work, although in that paper BLM and Recq15 were the focus.

Overall Critique:

The study of Tian et al. invokes a model in which zebrafish WRN has importance for bone development and growth, a finding consistent with another recently published study in DNA repair. The apparent advance here is that short stature homeobox gene SHOX expression is regulated by WRN, which may be a causative factor for *wrn*^{-/-} related phenotypes; however, further experimental studies might have more strongly supported this conclusion. Furthermore, the authors of the current study propose that WRN regulates SHOX gene expression via resolution of promoter G-quadruplexes of the SHOX gene; however, it is unclear how specific this function of WRN truly is, given that there are a number of G4-resolving helicases in humans. Extrapolation of the findings from zebrafish to human remains uncertain and likely would be more complex in the higher order vertebrate. It remains to be fully

understood if the dysregulation of SHOX due to loss of WRN G4 resolvase activity underlies the short stature in humans with WS, or if the molecular-genetic basis is more complex.

Dear Editors and Reviewers,

Thank you very much for your comments concerning our manuscript entitled “NCOMMS-21-39263 WRN promotes bone development and growth by unwinding SHOX-G-quadruplexes via its helicase activity in Werner Syndrome”. Your comments, suggestions, and requirements for changes in the manuscript have been very valuable and helpful for revising and improving the manuscript. We have carefully studied all comments and have made corrections which we hope answers the requirements set during the revision. Revised parts of the manuscript are marked in red text. Our point-to-point responses to the reviewers’ comments are shown below.

REVIEWER COMMENTS

Reviewer #1 (Remarks to the Author):

This work by Tian et al., focuses on Werner syndrome (WS), a monogenic autosomal recessive disorder caused by mutations of WRN gene. Patients affected by WS are characterized by short stature alongside to other phenotypes. Using a zebrafish *wrn*^{-/-} mutant as a model for WS herein, the authors unveil that *wrn* is a regulator of bone development and growth through transcriptional regulation of *Shox* gene. Moreover, they provide evidence that *Wrn* protein triggers this function by unwinding *Shox*-g –quadruplex via its helicase domain.

We sincerely appreciate reviewer 1’s all the valuable comments which have greatly improved our manuscript.

Comments:

1. This zebrafish mutant appears to be a good WS model, and the study focuses exclusively on the skeletal phenotype in order to unveil the underlying molecular mechanism(s). Do the authors have any knowledge if this zebrafish *wrn*^{-/-} mutant features any other phenotypes known to be associated to WS? (e.g. premature replication senescence; increased chromosome instability; increased apoptosis; skin defects, like thinning). These would be valuable information since the mouse homolog mouse mutant does not phenocopy so effectively the WS.

Response: Thank you for the comments on our *wrn*^{-/-} zebrafish model. As pointed out by the reviewer, the WS zebrafish model is suitable for evaluating skeletal phenotypes related to short stature. In addition to this feature, *wrn*^{-/-} zebrafish show a shorter lifespan compared to wildtype zebrafish, as demonstrated previously (1). They also showed increased cellular senescence and apoptosis, which was examined in a chondrocyte context, and fat loss in the abdomen of adult zebrafish.

Ref 1. Shin U et al. Large-scale generation and phenotypic characterization of zebrafish CRISPR mutants of DNA repair genes. *DNA Repair (Amst)*.**107**, 103713 (2021)

2. Figure 2, Panels o-v showing the extent of cartilage formation and proliferative status between WT and *Wrn*^{-/-} mutant reveal less cartilage differentiation as well less proliferative activity in the mutant. How the authors comment this observation? Proliferation and differentiation are usually two mutually exclusive events during development.

Response: We appreciate your valuable comments. Previous studies of WS showed that loss of WRN causes DNA damage (2) that may inhibit differentiation (3, 4) and proliferation (5). Zhang et al detected increased ROS levels, enhanced γ H2AX expression levels, and skeletal growth retardation in parathyroid hormone-related peptide (PTHrP) knock-in mice (3). Additionally, Li et al observed that increased apoptosis caused bone loss and inhibited osteoblast differentiation in *Atg7* mice (4). Based on these previous studies, we examined the expression level of γ H2AX, a DNA damage marker. As shown in Fig. a below (summarized in Supplementary Fig. 2b in the revised manuscript) shown below, the expression level of γ H2AX was increased in *wrn*^{-/-} zebrafish compared with that in the wildtype, indicating that loss of *wrn* led to DNA damage. Additionally, we evaluated apoptosis in the vertebrate regions by detecting *in situ* apoptosis (Invitrogen, C10618). As shown in Fig. b below (summarized in Supplementary Fig. 2a in the revised manuscript), more apoptosis signals were found in *wrn*^{-/-} zebrafish compared with that in the wildtype, which agreed with our previous results that less proliferative cells were observed in *wrn*^{-/-} zebrafish (summarized in Fig. 2p and r in the revised manuscript). Furthermore, it has been previously reported that DNA damage drives bone aging and inhibits bone differentiation, indicating that senescence plays a critical role in bone differentiation (5). Thus, we measured the expression of senescence markers, namely, *p53* and *p16*, using FISH analysis and observed the up-regulation of *p53* and *p16* in *wrn*^{-/-} zebrafish (Fig. c shown below, summarized in Fig. 8k-m in the revised manuscript). We also detected more senescent chondrocytes in both sh-WRN hESCs and sh-WRN hMSC compared with those in the CTR groups using flow cytometry analysis (Fig. e, f, i, and j shown below, summarized in Fig. 8e, f, i, and j), indicating that loss of *WRN/wrn* could lead to chondrocyte senescence. More importantly, overexpression of *SHOX/shox* (name as the rescue group) in cells and zebrafish promoted proliferation (data please refer to Fig. 7l, n, p, s, u, and w in the revised manuscript) and prevented the senescence phenotype (Fig. g, k, l, and m shown below, summarized in Supplementary Fig. 8g, k, l, and m) compared with that in sh-WRN hESCs and sh-WRN hMSCs. Based on these results, WRN/*wrn* loss-induced apoptosis and senescence is one possible explanation for why both differentiation and proliferation were inhibited.

We also added the related contents in the manuscript: Page 8 from line 233 to line 239, page 24 from line 597 to line 615 in red font in results section. And page 31 line 752 to line 760 in the discussion section.

Ref 2: Li SC et al. WRN, the protein deficient in Werner syndrome, plays a critical structural role in optimizing DNA repair. *Aging Cell*.**2**, 191-199 (2003)

Ref 3: Zhang Y et al. DNA damage checkpoint pathway modulates the regulation of skeletal growth and osteoblastic bone formation by parathyroid hormone-related peptide. *Int J Biol Sci*.**14**, 508-517 (2018)

Ref 4: Li HX et al. Defective autophagy in osteoblasts induces endoplasmic reticulum stress and causes remarkable bone loss. *Autophagy*.**14**, 1726-1741 (2018)

Ref 5: Chen Q et al. DNA-damage drives accelerated bone aging via an NF-kappaB-dependent

Reviewer 1---Question 2

3. Figure 3 and Sup Fig.2, show cartilage formation, under the material and methods section the authors when describe the chondrogenic procedure state that chondrocyte differentiation from hESCs was performed following a previously published protocol (32), referring to Bianco P et al. Skeletal stem cells. Development 142 1023-1027 (2015). This paper is a review and importantly, Bianco and Robey in the context of chondrogenic differentiation assay procedure to follow state that “Cartilage formation in micro-mass or pellet cultures is more reliable, as it rests on ultimate histological proof of genuine cartilage”. The authors have performed their chondrogenic assay on cell-monolayer instead.....

Response: Thank you for the comment on the reference and protocol. We understand the reviewer’s concern regarding the cell-monolayer method, and we apologize for the error in the reference. We followed the protocol described in the article, “Directed Differentiation of Human Embryonic Stem Cells Towards Chondrocytes” (Rachel AO et al. *Nature Biotechnology*, 2010) (reference number is 35 in the revised manuscript, previously reference number is 52). The advantage of this model (Fig. a shown below, summarized in Fig.3a in the revised manuscript) is that it fully recapitulates the whole chondrogenic development process from the pluripotent stage to the mesoderm stage; and chondrocyte stage. Pluripotent hESCs (stage 1, day 0) were directed towards a primitive streak-mesendoderm population (stage 2, day 4), after which differentiation proceeded to a mesoderm population (stage 3, day 9), and finally towards chondrocytes (stage 4, day 14), which is useful for examining the complete process of chondrocyte differentiation in the context of WS. We also used human mesenchymal stem cells as our second cell model (Fig. b shown below, summarized in Supplementary Fig.3a in the revised manuscript); chondrogenesis in the hMSC model was highly similar to that in the hESC model from stages 3 to 4. We performed micromass cell culture for chondrogenic differentiation of hMSCs. The micromass cell culture provides a three-dimensional environment that allows the cells to grow in aggregates, which is suitable for generating a high cell density and cell-cell interactions to better induce chondrogenesis (6)

Ref 6: Carolina et al. Micromass cultures are effective for differentiation of human amniotic fluid stem cells into chondrocytes. *Clinics (Sao Paulo)*.**73**, e268 (2018)

Reviewer 1 ---Question 3

4. Apoptosis is known to be a crucial step during endochondral ossification, additionally misregulation of genes involved in apoptosis has been reported in mice lacking a functional *Wrn* protein and *Wrn* knockdown inhibits proliferation while promoting apoptosis. However, the authors have completely ignored this aspect. Therefore, investigation on apoptosis would be desirable in order to rule out that potential cell competent to specify towards the chondrogenic lineage are negatively selected by apoptosis.

Response: Thank you for pointing out the essential link between the loss of WRN and the induction of apoptosis and inhibition of proliferation. With regards to bone development, the apoptosis is thought to play a critical role during skeletal development (7). Therefore, we evaluated apoptosis in both cells and zebrafish at different bone developmental stages to exclude the possibility that chondrocytes are negatively selected by apoptosis. As shown by poly-caspases and PI staining, we found that many apoptotic chondrocytes at an early stage both in shWRN1# hESCs and shWRN1# hMSCs (Fig. a-b shown below, summarized in Supplementary Fig. 4a-b). We noted the accumulation of apoptotic cells began appearing at stage 2, which is the premesoderm stage in the hESC model, with more apoptotic chondrocytes accumulated at stage 4 (Fig. a shown below, summarized in Supplementary Fig. 4a). Similarly, the number of apoptotic chondrocytes increased during bone development in the hMSC model (Fig. b shown below, summarized in Supplementary Fig. 4b in the revised manuscript). In agreement with the staining results, the apoptosis markers *BCL2* and *CASPASE 8* showed enhanced expression in shWRN1# hESCs and shWRN1# hMSCs compared to CTR hESCs or hMSCs (Fig. c-f shown below, summarized in Supplementary Fig. 4c-f in the revised manuscript). Moreover, a larger number of apoptosis signals were found in *wrn*^{-/-} zebrafish from 3 dpf to 14 dpf compared with that in the wildtype in the vertebrate regions (Fig. g shown below, summarized in Supplementary Fig.2a in the revised manuscript). Collectively, these results suggest that loss of WRN leads to early apoptosis and exclude the possibility that chondrocytes are negatively selected by apoptosis.

We also added the related contents in the manuscript: Page 8 from line 233 to line 237, page 11 from line 342 to line 345 in results section.

Ref 7: Whit A et al. Endochondral ossification: A delicate balance between growth and mineralization. *Curr Biol.***11**, R589-591

Reviewer 1---Question 4

5. Supp. Figure 4, Panel g, illustrates FISH analysis of *shox* expression in WT zebrafish and *wrn*^{-/-} mutant zebrafish 7 dpf. It would be of interest to analyze the expression at other time points, such as dpf 3 and dpf 14 and in parallel to the expression of *wrn*. This profiling would provide better and complete information regarding the co-expression of these two genes during chondrogenesis.

Response: Thank you for the suggestions. We have added more time points for *shox* expression in both the wildtype and *wrn*^{-/-} zebrafish, as suggested. As shown below (summarized in Supplementary Fig. 7h-j), *wrn*, *shox* and BrdU signals accumulated in wildtype zebrafish from 3 to 14 dpf according to FISH analysis; this accumulation was not observed in *wrn*^{-/-} zebrafish, indicating that *wrn* and *shox* were co-expressed during chondrogenesis.

We also added the related contents in the manuscript: Page 17 from line 456 to line 459.

Reviewer 1---Question 5

6. Figure 7, panels k-m and u-w show FISH analysis for Col2a1a and col10a1a expression combined with BrdU staining. The authors comment Shox1 overexpression in *wrn*^{-/-} zebrafish “significantly enhanced” the expression of these genes as well the proliferation activity. It is hard to believe that a staining can be quantified as significant..... I would suggest a qRT in order to establish if the outcome is significant. Moreover, in these panels it is hard to appreciate Sox9 and Col 10a1a staining, whereas it is possible for Col2a1a.

Response: Thank you for the comments. We have added the qRT-PCR results. These results are shown below and summarized in Fig.7k and r. These results are in agreement with our FISH analysis in Fig. 7, showing that the loss of *wrn* led to the decreased expression of *sox9a*, *col2a1a*, and *col10a1a*. Moreover, overexpression of *shox* promoted these genes expression in *wrn*^{-/-} zebrafish. In the meantime, we have corrected our description in the manuscript: Page 22 from line 560 to line 564.

Reviewer 1---Question 6

Reviewer #2 (Remarks to the Author):

Werner syndrome (WS) is a rare form of accelerated ageing disease with clinical characteristics resembling those of normal ageing. WS is caused by multiple mutations in the gene encoding the Werner DNA helicase (WRN), but the underlying mechanisms driving short stature in WS patients remains elusive. In this paper, Tian et al., investigated the role of WRN gene in regulating of chondrogenesis and the underlying mechanisms using zebrafish as an *in vivo* model. The authors found that WRN deficiency results in the inhibition of bone growth and short stature. Moreover, the authors, using *in vitro* cultured human embryonic stem cells (hESCs) and human mesenchymal stem cells (hMSCs), demonstrated that WRN deficiency impairs cartilage development. The authors also performed RNA-seq and ChIP-seq and showed that the SHOX (short-stature homeobox) gene is a direct target of WRN. WRN regulated SHOX expression by facilitating gene transcription. In addition, the authors demonstrated that manipulating the expression of SHOX is sufficient to mimic or rescue chondrogenesis and bone formation in WT or WRN knockdown models. The authors should be commended on their well-conducted study and engaging manuscript, which contains an impressive amount of data that are interpreted to support some interesting conclusions. I have the following points for the authors to address:

We are grateful to reviewer 2 for his/her effort in reviewing our manuscript and gave us all the helpful comments.

Major comments

It has been recognized that DNA damage, cellular senescence, telomere attrition, and impaired autophagy/mitophagy resulting from mutations in WRN explain the majority of the clinical features of WS. It is somewhat disappointing that these hallmark changes were not evaluated in the testing model systems. Particularly, from the BrdU labeling (Figure 2) and the RNA-seq data (Figure 5), cell proliferation/cell growth-associated genes are among the most significantly downregulated pathways in the WRN deficiency cells (vs. WT cells), suggesting that cellular senescence is likely involved in WRN deficiency-caused bone development and growth defects. A previous report, which is closely related to the current study, showed that stem/progenitor cells in metaphysis of long bone during rapid growth period are highly proliferative but undergo progressive cell senescence during late puberty when bone growth slows down (Nat. Commun. 2017; 8:1312), indicating that cellular senescence is a key mechanism to control bone growth. Therefore, it would be interesting for the authors to assess whether loss of WRN causes accumulation of senescent cells in the zebrafish and the *in vitro* stem cells culture model systems. It is also interesting to test whether overexpressing SHOX prevented the senescence phenotype.

Response: We appreciate the valuable comments. Depletion of WRN has been reported to cause accumulation of senescence-associated- β -galactosidase (SA- β -gal) positive hMSCs (8). To determine whether the loss of WRN lead to the accumulation of senescent chondrocytes, we performed senescence assays both *in vitro* and *in vivo*. We first examined cellular senescence using

senescence flow cytometry analysis (Invitrogen, C10840) by labeling SA- β -gal positive cells. Gating strategy is shown below in Fig. a (hESC model, summarized in Fig. 8a) and Fig. e (hMSC model, summarized in Fig. 8e). Flow cytometry analysis revealed an increased number of SA- β -gal-positive cells (P3), with 29.6% senescent cells detected in shWRN1# hESCs and 30.1% in shWRN1# hMSCs; only 0.2% of hMSCs were senescent in the control groups and 0.0% in control hESCs (Fig. b, c, f, and g shown below, summarized in Fig. 8b, c, f, and g). After overexpression of *SHOX* in shWRN1# hESCs and shWRN1# hMSCs, the number of senescent cells decreased to 13.5% and 14.1%, respectively (Fig. d and h shown below, summarized in Fig. 8d and h). We also evaluated the expression of two senescence markers, P53 and P16, both in stem cells and zebrafish models. The mRNA expression of *P53* and *P16* increased in shWRN1# hESCs and shWRN1# hMSCs compared with that in CTR cells; overexpression of *SHOX* decreased the expression of these two genes (Fig.k-n, summarized in Fig. 8k-n). Similar results were observed in the zebrafish model at different development stages (Fig. o-q, summarized in Fig. 8o-q). Together, these results indicate that loss of *WRN/wrn* caused cellular senescence, whereas overexpression of *SHOX* prevented the senescence phenotype.

The related contents were added in the revised manuscript: Page 24 from line 597 to line 615 in results section. Page 33 from line 840 to 843 in the discussion section.

Ref 8. Zhang WQ et al. Aging stem cells. A Werner syndrome stem cell model unveils heterochromatin alterations as a driver of human aging. *Science* **348**. 1160-1163 (2015)

Reviewer 2--Question 1

a Gating strategy for senescence positive chondrocytes (hESC model)

e Gating strategy for senescence positive chondrocytes (hMSC model)

Minor points

1. The authors should give an overview in the Introduction section on the current understanding of the cellular and molecular mechanisms that result in the accelerated aging of WS patients.

Response: Thanks for the kind suggestions, we have added the related contents on page 3 from line 79 to line 83 in the introduction section.

2. The global WRN deficiency models was used in the study. To better demonstrate WRN primarily functions in early-stage chondrocyte/MSCs, it would be helpful to conduct double-immunostaining of WRN/SHOX with chondrocyte/MSC markers in the model system.

Response: Thank you for the suggestions. We have performed the double-immunostaining of WRN/SHOX in the MSC cell model. As results shown below, WRN and SHOX showed a highly similar expression pattern during chondrogenesis.

The related contents were added in the revised manuscript: Page 19 from line 484 to line 487 in results section.

2. In Figure 5d, f, it seems SHOX is not the most down-regulated gene according to the z-score and the heatmap. However, SHOX is the most significantly downregulated genes in Figure 5g, h. Can the authors explain the discrepancy?

Response: Thank you for your question. One possible explanation is that the three biological replicates in the control group were less than ideal, and this bias may have affected differential gene expression analysis of the WRN and control groups. However, our qPCR analysis and zebrafish FISH analysis clearly showed that loss of WRN/wrn led to downregulation of SHOX/shox. Additionally, we checked the RNA-seq dataset from Tu et al (9) and found that the SHOX gene is among the top 5 upregulated genes in WS-corrected fibroblasts, indicating that SHOX is a crucial gene regulated by WRN.

Ref 9: Tu JJ et al. Genetic correction of Werner syndrome gene reveals impaired pro-angiogenic function and HGF insufficiency in mesenchymal stem cells. *Aging Cell* **19**. e13116 (2020)

3. In the ChIP-qRT-PCR data (Figure 8g, h), it would be better if the authors show the PCR gel images with negative and positive controls.

Response: Thank you for the suggestion. We have now added the negative and positive controls (Previous in Fig. 8g and h, now summarized in Fig. 9f and g). Additionally, we added the negative and positive controls of suppl. Fig. 10a and b (now summarized in Supplementary Fig. 13e and f). All the new PCR gels results were summarized in Supplementary Fig. 13a-d in the revised manuscript.

We used IgG as the negative control. We designed RNA polymerase II binding regions in human GAPDH promoter primers (192 bp) as the positive control. Anti-RNA polymerase II antibody was purchased from Thermo (26157).

The results were shown below and also summarized in Supplementary. Fig. 13 a-d, and the related contents were added in the revised manuscript: Page 27 from line 665 to 667.

Forward Primer (5'---3'): CCC AAA GTC CTC CTG TTT CA

Reverse Primer (5'---3'): TTT TCC GCA GCC GCC TGG TTC A

Reviewer 2---Question 5

2% Agrose Gel

DNA Marker: Invitrogen 100 bp DNA ladder (Cat. 15628019)

Fig.8g and h now summarized in Fig. 9f and g. Supple. Fig.10a and b now summarized in Supple. Fig. 13 e and f.

Reviewer #3 (Remarks to the Author):

Summary:

Skeletal dysplasia and short stature are clinical phenotypes of considerable complexity with the large number of genes and spectrum of clinical disorders. SHOX is a key regulator of chondrocyte biology, as reflected by the multiple number of target genes that it regulates. While zebrafish serve as a useful model for whole body characterization of genes regulating height and skeletal components, the pathways are quite complex. Moreover, relating skeletal phenotypes of zebrafish to human and tissue-specific gene regulation is formidable. In the current work, the authors have investigated the hypothesis that the WRN helicase-nuclease implicated in the premature aging Werner syndrome regulates expression of SHOX via its G4 resolvase activity on SHOX-promoter associated G-quadruplexes.

The authors document shortened body length and bone abnormalities in *wrn*^{-/-} zebrafish. From these studies they transition to human embryonic stem cells and human mesenchymal stem cells to examine the role of WRN in chondrogenesis at the cellular level. However, the direct connection of their observations remain elusive to the zebrafish model findings. To their credit, the authors assess if human WRN mRNAs injected into *wrn*^{-/-} mutant zebrafish at the one-cell stage on regulation of expression of key genes in the chondrogenesis pathway. By this analysis, they find that SHOX expression is regulated by WRN during chondrocyte differentiation, and this is largely dependent on WRN helicase activity. SHOX expression on stature was further examined by microinjection of human SHOX mRNA into zebrafish embryos at the one-cell stage and embryo length was correlated with bone growth/development expression. Finally, the authors determine that WRN unwinds promoter-associated G4, thereby regulating its expression.

We truly appreciate reviewer 3 for his/her great effort reviewing our manuscript and all the valuable comments, which helped us improved the manuscript a lot.

Critical Comments:

SHOX signaling is notably complex, and there is still only limited understanding of the pleiotropic effects of SHOX deficiency in humans. Therefore, the utilization of model genetic organisms with conserved genes and translational value is appreciated. Zebrafish is one such model for studying chondrogenesis, but has its limitations that are apparent at the organismal and tissue levels.

A useful tool in the current study might have been G-quadruplex ligands to modulate SHOX promoter function and interrogate WRN involvement. This experimental approach might have been applied in the *in vivo* model.

Response: We truly appreciate your valuable suggestions. We synthesized an oligonucleotide containing the zebrafish G4 region (*z-shox-g4*), as well as two oligonucleotides without *z-shox-g4*. We tested the formation of *z-shox-G4* *in vitro* in a ThT assay. The fluorescent intensity (F/F₀) represented the presence of G4s (a value of more than 20 is typically regarded to indicate G4 formation. Human C-MYC G4 pu18 was used as the positive control, and the pu18 mutant was used as the negative control according to previous report (10)). Fig. a (summarized in Supplementary Fig. 13j in the revised manuscript) showed that the formation *z-shox-G4s* *in vitro*, which were cloned

into pGL3SV40 vectors (Promega). The dual-luciferase activity assay was conducted by transfecting z-shox-g4 or z-shox-non-g4 and full-length *wrn* together into sh-WRN-293T cells (the knockdown efficiency was confirmed as Fig. b shown (summarized in Supplementary Fig. 13i in the revised manuscript)). We found that full-length *wrn* increased *shox* expression more significantly compared with that in the z-shox-non-g4 groups, indicating that G4s are crucial for gene transcription *in vitro* (Fig. c (summarized in Fig. 9i in the revised manuscript)).

However, the actual effect of these G4s in a cellular and genomic natural environment may be different. An important question is whether G4s are truly involved in transcriptional regulation *in vivo* during bone development. To answer this question, we disrupted G4 ligands by microinjecting z-*shox* anti-sense oligonucleotides (ASO) complementary to the selected G4s (Fig. d (summarized in Fig. 9j in the revised manuscript)) into *wrn*^{-/-} zebrafish. We injected 10 pg-ASO/embryos or controls (an oligonucleotide that did not anneal to the zebrafish genome was used as the control (CTRL) (11)) into one-cell staged zebrafish embryos (a previous report had confirmed this optimal concentration by testing the survival of specimens until 48 hpf (11)). After confirming the efficiency of anti-sense oligonucleotides (ASO) (Fig. e (summarized in Supplementary Fig. 13k in the revised manuscript)), we performed RT-qPCR, which revealed that *shox* expression increased during development (Fig. f (summarized in Fig. 9k in the revised manuscript)). We also microinjected h-WRN and h-WRN helicase mutant mRNAs into *wrn*^{-/-} zebrafish. Overexpression of h-full-length-WRN mRNA promoted *shox* expression, whereas h-WRN helicase mutant did not, indicating that G4s are crucial for regulating gene expression, and WRN regulates *SHOX* expression by unwinding *SHOX*-G4s via its helicase activity.

The related contents were added in the revised manuscript: Page 27 from line 679 to line 700 in results section.

Ref 10: Renaud FA et al. Thioflavin T as a fluorescent light-up probe for G4 formation. *Nucleic Acids Res.***42**. e65 (2014)

Ref 11: Aldana PD et al. G-quadruplexes as novel cis-elements controlling transcription during embryonic development. *Nucleic Acids Res.***44**. 4163-4173 (2016)

Reviewer 3---Question 1

Specificity of WRN's effect was not addressed. One wonders if the sequence-related BLM helicase which has also been reported to regulate gene expression via its action on G4 promoters would affect the observed phenotypes via SHOX.

In my mind, it remains unclear if the effect of WRN is specific to regulation of SHOX or if other gene regulatory factors are implicated in genetic determination of height and bone homeostasis are involved, that are also directly regulated by WRN or indirectly. While the study suggests WRN is involved, a direct causal relational and mechanism is not established.

Response: Considering the reviewer's suggestion, we evaluated the BLM expression pattern during cartilage development both *in vitro* and *in vivo*. As shown below (summarized in Supplementary Fig. 5 in the revised manuscript), there was no significant difference in the expression of *BLM* between CTR cells and shWRN1# and shWRN2# cell in both the hESC and hMSC models. Additionally, we measured the expression of *wrn* and *blm* at 7 dpf and 14 dpf in zebrafish; *blm* signals were still detected in *wrn*^{-/-} zebrafish, indicating that *BLM/blm* might not be crucial for regulating chondrogenesis and body height in WS. Ken et al suggested that functional deletion of either WRN or BLM cannot be compensated by the other protein (or other Rec Q proteins), indicating that WRN and BLM play different roles in cells (12,13).

We added related contents in the revised manuscript on page 13 from line 378 to line 384 in red font in results section, and page 31 from line 783 to line 788 in red font in the discussion section.

Ref 12: Man FJ et al. The human WRN and BLM RecQ helicases differentially regulate cell proliferation and survival after chemotherapeutic DNA damage. *Cancer Res.* **70**, 6548-6555 (2010)

Ref 13: Ken K. Structural mechanisms of human RecQ helicases WRN and BLM. *Front Genet.* **5**, 366 (2014)

Reviewer3---Question 2

Assessment of WRN connected SHOX gene regulation on other tissues of zebrafish could be performed to provide a better impression of the broadness of the gene regulatory network.

As shown by the authors, 380 genes were found to be regulated by WRN in chondrocyte homeostasis, 116 up-regulated and 264 down-regulated. Given the large number, it may be that multiple genes co-regulated with SHOX or even directly regulated by SHOX are also implicated, as suggested. It would have been informative and supportive of their model if the authors had examined in greater detail the mechanism whereby some of the top candidate involved in chondrogenesis, as evidenced by previous studies, are affected and if their regulation is relevant to the phenotypes of *wrn*^{-/-} zebrafish model in this work. This would apply to SOX9, SOX5, SOX6 (The transcription factors SOX9 and SOX5/SOX6 cooperate genome-wide through super-enhancers to drive chondrogenesis - PubMed (nih.gov)). The gene regulatory networks of chondrocytes are quite complex, and super enhancers as well as histone modifications and chromatin accessibility come into play. It has been proposed that multiple transcription factors are involved in chondrocyte biology. While the evidence from the current study suggests that WRN regulation of SHOX is among these, it is difficult to comprehend the molecular-genetic mode of action in the context of the multiplicity of transcription factors coming into play.

Response: Thank you for these suggestions and comments. We agree with the reviewer that the chondrocyte biology and regulators are complex and depends on several factors. SHOX is one of the critical factors in regulating bone development, particularly in chondrogenesis. Considering the reviewer's suggestions, we overexpressed *shox* by microinjecting *shox* at the one-cell stage and collected non-treated and treated *wrn*^{-/-} zebrafish at different time points. As shown below (summarized in Supplementary Fig.11), overexpression of *shox* promoted the expression of *sox9b*, *sox5*, and *sox6* compared with that in *wrn*^{-/-} zebrafish. *sox9* has two isoforms in zebrafish, namely *sox9a* and *sox9b*. We previously found that overexpression of *shox* promoted *sox9a*, *col2a1a*, and *coll10a1a* expression in *wrn*^{-/-} zebrafish. (data please refers to Fig. 7k-p and Fig. 7r-w)). Taken together, these data suggested that *shox* co-regulated with *sox9a*, *sox9b*, *sox5*, and *sox6* in chondrogenesis. Our study is agreed with previous report that SHOX can physically interact with various proteins, such as SOX9, SOX5, and SOX6, to promote bone development (14). Additionally, Mirial et al reported that SHOX directly interacts with SOX5 and SOX6 and that mutations in SHOX inhibit the interactions (15). Moreover, SHOX, SOX9, SOX5, and SOX6 were coexpressed in 18- and 32-week human fetal growth plates, indicating that SHOX coregulate these genes during skeletal development and determination of the body stature (15). However, whether SHOX can directly interact with SOX9 is unclear. *Shox* can co-regulate with *Sox9* (*Sox9* plays a crucial role in chondrogenesis by promoting the expression of cartilage matrix genes, such as *Col2a1a* and *Aggrecan*, which may further promote skeletal growth and development) in promoting chicken limb growth (16).

We added related contents in the revised manuscript on page 22 from line 564 to line 569 in red font in results section, and page 33 from line 836 to line 840 in red font in the discussion part.

Ref 14: Aza-Carmona M et al. SHOX interacts with the chondrogenic transcription factor SOX5 and SOX6 to activate the aggrecan enhancer. *Hum Mol Genet.***20**, 1547-1559 (2011)

Ref 15: Marchini A et al. A track record on SHOX: from basic research to complex models and therapy. *Endocr Rev.***37**, 417-448 (2016)

Ref 16: Tiecke E et al. Expression of the short stature homeobox gene *shox* is restricted by proximal and distal signals in chick limb buds and affects the length of skeletal elements. *Dev Biol.* **298**, 585-596 (2006)

Reviewer 3---Question 3

Based on their results, the authors suggest that WRN controls bone growth and development via its regulation of SHOX. Are there other phenotypes beyond bone-related affected by WRN status and are these also mediated by SHOX? The complexity of WRN's genetic linkage to range of clinical features of accelerated aging. This is compounded by the extensive data that at least in vitro (biochemical), cell-based models, and genetic models, WRN plays roles in transcriptional regulation, DNA repair, replication stress response, genomic stability. So is this an oversimplification that WRN's involvement in controlling bone metabolism in zebrafish is mediated by its effect on SHOX, and there is no consideration of its pan-wide functions in multiple cell types and tissues?

Response: Thank you for raising this important point. Considering the reviewer's question, qRT-PCR was performed to evaluate the expression patterns of *wrn* and *shox* at 40 dpf both in wildtype and *wrn*^{-/-} zebrafish (most zebrafish tissues and organs have formed at this stage). As shown below (also summarized in Supplementary Fig.7g-h in the revised manuscript), the *wrn* gene was universally expressed in the wildtype zebrafish, and the *shox* gene mainly expressed in the bones (vertebrate bone and craniofacial bone) and slightly in the brain, indicating that *shox* mainly participates in bone tissue related activities, in agreement with a previous report (17). Although WRN may function in other cell types and tissues, we mainly evaluated the WRN-SHOX interaction in bone development. Our data clearly suggest that WRN-SHOX regulates chondrogenesis and zebrafish body length.

We added related contents in the revised manuscript on page 17 from line 450 to line 456 in red font in results section.

Ref 17. Hoffmann S et al. Identification and Tissue-Specific characterization of novel SHOX-regulated genes in zebrafish highlights SOX family members among other genes. *Front Genet.* **12**, 688808 (2021)

Another layer of potential complexity is added with the idea that WRN's key molecular activity of G4 resolution in the promoter of SHOX underlies the growth/bone phenotypes may be overly simplified. WRN, along with other G4-resolving helicases (e.g., BLM) is thought to play a role in regulation of expression of many genes with G4-laden promoters, not to mention its effect on G4 in other chromosomal regions (e.g., telomeres). I'm not sure all the phenotypes related to chondrocyte biology hinge on WRN-catalyzed G4 resolution of SHOX promoter DNA sequence elements. This seems to be the driving conclusion of the paper, and I am not sure that other avenues have been explored which potentially contribute to the observations made for *wrn*^{-/-} zebrafish.

Response: Thank you for raising the important point. It has previously been reported that WRN preferentially binds to a “bubble” structure, such as duplex DNA and G4 quadruplex, where it functions as a helicase to open these structures to promote DNA transcription, duplication, and replication (18). Gene promoters correlate highly with gene expression. As described in our response to Question 1, we synthesized a zebrafish G4 region which was cloned into the pGL3SV40 vector. We co-transfected z-*shox*-g4- pGL3SV40 with the zebrafish full-length *wrn* plasmid into shWRN-293T cells. These cells showed increased *shox* expression compared with that in the control cells (Fig. a-b, summarized in Supplementary Fig. 13j and Fig. 9i). To examine whether WRN preferentially resolves G4 structures, we synthesized two regions in the promoter without zebrafish G4 quadruplexes and performed the dual luciferase assays again. As shown below (Fig. a-b, summarized in Supplementary Fig. 13j and Fig. 9i), we found that *wrn* could promote *shox* expression significantly especially if there was a G4 structure in the promoter, indicating that WRN helicase is more sensitive to recognize a G4 structure and open it for further gene transcription and expression. X-ray crystallographic analysis has shown that WRN helicase preferred to bind to these “bubble structures” due to its unique winged-helix domain of WRN protein (19). In the meantime, we noted that the disruption of z-*shox*-g4 ligands enhanced chondrogenesis, similar effect was observed when microinjecting h-full-length-WRN mRNAs, indicating that WRN-catalyzed SHOX-G4-promoter is crucial in regulating chondrogenesis (Fig. c-d, summarized in Fig. 9l-n). We added related contents in the revised manuscript on page 27 from line 679 to line 704 in red font in results section, and page 33 from line 857 to line 865 in discussion section.

Ref 18: Michael Fry. The Werner syndrome helicase-nuclease—one protein, many mysteries. *Sci Aging Knowledge Environ.* **2002**, re2 (2002)

Ref 19: Ken K et al. Structural basis for DNA strand separation by the unconventional winged-helix domain of RecQ helicase WRN structure. *Structure.* **18**, 177-187 (2010)

A recently published study by Shin et al., DNA Repair (2021) Large-scale generation and phenotypic characterization of zebrafish CRISPR mutants of DNA repair genes; Large-scale generation and phenotypic characterization of zebrafish CRISPR mutants of DNA repair genes - PubMed (nih.gov) reported that *wrn*^{-/-} zebrafish. Like *mre11*^{-/-} zebrafish and certain other DNA repair mutants displayed reduced growth and development during the juvenile stage. Interestingly, both *mre11acu56/cu56* and *wrnacu64/cu64* failed to survive to 60 dpf. The similar phenotype raises the question if the genetic defect is attributed to unusual G4 metabolism or more generically a DNA repair defect such as double-strand break repair in which both WRN and MRE11 are implicated. It would have been of interest if the authors of the current study had determined what phenotypes and at what developmental stage were observed in *wrn*^{-/-} zebrafish that had been exposed to a G4-binding ligand. If the temporal appearance or severity of the observed phenotypes were G4-inducible, then this would support their model.

Response: We truly appreciate your valuable comments. Unusual G4 metabolism is associated with DNA repair defects, DNA damage etc (20). We examined the expression of γ H2AX in both wildtype and *wrn*^{-/-} zebrafish. As shown in Fig. a below (also summarized in Supplementary Fig. 2b in the revised manuscript), the expression of γ H2AX in the vertebrate regions increased in *wrn*^{-/-} zebrafish as developmental stages proceed. As described in our response to Question 1, we designed an antisense oligonucleotide (ASO) targeting zebrafish *shox*-G4 ligands and microinjected this ASO at the one-cell stage (Fig. b, summarized in Fig. 9j in the revised manuscript). After confirming the efficiency of *shox*-g4-ASOs (Fig. c, summarized in Supplementary Fig. 13k in the revised manuscript), we evaluated *shox* expression at different developmental stages, which was found to be increasing over time compared with that in the control groups (Fig. d, summarized in Fig. 9k in the revised manuscript). Additionally, we microinjected the human full-length WRN mRNA and the human helicase mutant mRNA respectively (Fig. d, summarized in Fig. 9k in the revised manuscript). Compared to ASO group, overexpression of human WRN mRNA in *wrn*^{-/-} zebrafish increased *shox* gene expression, whereas the human helicase mutant did not. Together, these data demonstrate that G4 structures are present at an early developmental stage, and that their formation is inducible.

We added related contents in the revised manuscript on page 27 from line 689 to line 704 in red font in results section.

Ref 20: Alessio DM, et al. DNA damage and genome instability by G-quadruplex ligands are mediated by R loops in human cancer cells. *Proc Natl Acad Sci U S A*. **116**, 816-825 (2019)

Reviewer 3---Question 6

Yokokura et al. (Front Endocrinology (Lausanne) 2017; The Short-Stature Homeobox-Containing Gene (*shox*/ *SHOX*) Is Required for the Regulation of Cell Proliferation and Bone Differentiation in Zebrafish Embryo and Human Mesenchymal Stem Cells - PubMed (nih.gov)) had assessed the phenotypes of morpholino oligo-mediated knockdown of zebrafish *shox*. Determination if *wrn* and *shox* genetically operate in the same pathway as predicted by the model proposed by the authors of the current study would have helped to strengthen the study.

Response: Thank you for your comments. As we previously discussed, the expression pattern of *WRN* and *SHOX* were highly similar during chondrogenesis in both hESCs and hMSCs (Fig. a-d). Since chondrogenesis initiates from mesendoderm and the cells undergo mesenchymal condensation, proliferation and a series of other delicate events, they finally become chondrocytes (5). Our results revealed that *WRN* and *SHOX* proteins both appeared from the primitive-mesendoderm population (day 4, Fig. a-b) to the mesoderm population (day 9, Fig. a-b), and were also detected in chondrocytes (day 14, Fig. a-b). (Fig. a is summarized in Fig. 3a, Fig. b summarized in Supplementary Fig. 8a). Similar results were observed in the hMSC model (Fig. c (summarized in Supplementary Fig. 3b), Fig. d (summarized in Supplementary Fig. 9a)). These data indicate that both *WRN* and *SHOX* participate in the very early stage of chondrogenesis, as confirmed by immunostaining (Fig. e (summarized in Fig. 6a), Fig. f. (summarized in Supplementary Fig.9c)). Similar to the results observed in cells, *wrn* and *shox* were both expressed early from 3 to 14 dpf along the vertebrate regions in wildtype zebrafish but was not in *wrn*^{-/-} mutant zebrafish (Fig. g-i, summarized in Supplementary Fig.7h-j). Additionally, loss of *WRN* or *SHOX* may lead to decreased expression of Ki67, a proliferative marker, indicating that both genes are involved in proliferation (Fig. j (summarized in Fig. 3u), Fig. k (summarized in Fig. 6d), Fig. l (summarized in Supplementary Fig. 3o), and Fig. m (summarized in Supplementary Fig.9g)). Depletion of *WRN* or *SHOX* inhibited bone differentiation (data show in Fig. 3d-t (shWRN-hESC model), Supplementary Fig. 3d-n (shWRN-hMSC model), Supplementary Fig. 8c-n (shSHOX-hESC model), Supplementary Fig. 9d-f (shSHOX-hMSC model)). Collectively, these data indicate that *WRN* and *SHOX* genetically function in the same pathway regulating bone development.

Reviewer 3--Question 7

A 2007 paper by Xie et al in Genetics entitled Manipulating Mitotic Recombination in the Zebrafish through RecQ Helicases Manipulating Mitotic Recombination in the Zebrafish Embryo Through RecQ Helicases (nih.gov) might be relevant to the current work, although in that paper Blm and Recq15 were the focus.

Many thanks for sharing this interesting work with us, and it is helpful for our study.

Overall Critique:

The study of Tian et al. invokes a model in which zebrafish WRN has importance for bone development and growth, a finding consistent with another recently published study in DNA repair. The apparent advance here is that short stature homobox gene SHOX expression is regulated by WRN, which may be a causative factor for wrn^{-/-} related phenotypes; however, further experimental studies might have more strongly supported this conclusion. Furthermore, the authors of the current study propose that WRN regulates SHOX gene expression via resolution of promoter G-quadruplexes of the SHOX gene; however, it is unclear how specific this function of WRN truly is, given that there are a number of G4-resolving helicases in humans. Extrapolation of the findings from zebrafish to human remains uncertain and likely would be more complex in the higher order vertebrate. It remains to be fully understood if the dysregulation of SHOX due to loss of WRN G4 resolvase activity underlies the short stature in humans with WS, or if the molecular-genetic basis is more complex.

REVIEWER COMMENTS

Reviewer #1 (Remarks to the Author):

The authors have fully addressed my comments/questions by providing additional experimental data. Therefore, I consider the revised the manuscript suitable for acceptance.

Reviewer #2 (Remarks to the Author):

The authors have addressed my concerns. The authors should carefully check their reference list. In some references, authors first names and last names were misused. Please correct the format of the references to keep consistent with the journal requirement.

Reviewer #3 (Remarks to the Author):

The authors have addressed a number of critical points raised in my evaluation. Overall, the revised manuscript is improved. There are a couple lingering issues:

1. Expression patterns of *shox* and *wrn* were assessed, but the results from such analyses do not necessarily demonstrate that *wrn* and *shox* operate in the same pathway. In a true genetic sense with a genetic model, there is a requirement to assess the epigenetic relationship of two suspected genetic interacting factors, in this case *wrn* and *shox*.
2. A set of experiments which employ a small molecule G4-binding ligand that is amenable to cell-based or zebrafish model would have strengthened the working model that WRN's G4 resolvase activity of SHOX-associated G4 underlies WRN's involvement in bone development and growth. Stabilization of G4 by a G4-specific binding ligand would be expected to accentuate the bone-related phenotypes of a *wrn*^{-/-} mutant zebrafish model or characteristic phenotypes of human chondrocytes deficient in WRN.

Summary: The authors have addressed some points raised by my review, but it is still unclear to me if causal relationships among *wrn* status and G4-forming potential in *shox*, as it relates to bone phenotypes in the zebrafish or the human cellular phenotypes assayed, has been established.

REVIEWER COMMENTS

Reviewer #1 (Remarks to the Author):

The authors have fully addressed my comments/questions by providing additional experimental data. Therefore, I consider the revised the manuscript suitable for acceptance.

Response: We are grateful to reviewer 1 for his/her effort in reviewing our manuscript and the positive comments.

Reviewer #2 (Remarks to the Author):

The authors have addressed my concerns. The authors should carefully check their reference list. In some references, authors first names and last names were misused. Please correct the format of the references to keep consistent with the journal requirement.

Response: We truly appreciate reviewer 2's comments and suggestions, and we apologize for the errors in the reference list. We hereby carefully checked the reference format again and corrected the following references format:

42. Li, C. J. et al. Programmed cell senescence in skeleton during late puberty. *Nat Commun.***8**, 1312 (2017).

52. Wang, S. Y. et al. Ectopic hTERT expression facilitates reprogramming of fibroblasts derived from patients with Werner syndrome as a WS cellular model. *Cell Death Dis.***9**, 1-13 (2018).

Reviewer #3 (Remarks to the Author):

The authors have addressed a number of critical points raised in my evaluation. Overall, the revised manuscript is improved.

Response: Thank you for the reviewer 3's very important suggestions which have helped us improve the quality of our manuscript greatly.

There are a couple lingering issues:

1. Expression patterns of shox and wrn were assessed, but the results from such analyses do not necessarily demonstrate that wrn and shox operate in the same pathway. In a true genetic sense with a genetic model, there is a requirement to assess the epigenetic relationship of two suspected genetic interacting factors, in this case wrn and shox.

Response: Many thanks for the comments. In this study, we found that depletion of

WRN led to the downregulation of SHOX expression and inhibition of chondrogenesis. Additionally, we demonstrated that overexpression of *SHOX/shox* in WRN-KD cells or *wrn* mutant zebrafish could rescue cell proliferation and reduce cellular senescence, which are two crucial events and pathways in chondrogenesis (for data please refer to Fig. 2, Fig. 7, and Fig. 8). Collectively, our data indicated that WRN and SHOX work together, at least partially, in the same pathway to regulate chondrogenesis.

2. A set of experiments which employ a small molecule G4-binding ligand that is amenable to cell-based or zebrafish model would have strengthened the working model that WRN's G4 resolvase activity of SHOX-associated G4 underlies WRN's involvement in bone development and growth. Stabilization of G4 by a G4-specific binding ligand would be expected to accentuate the bone-related phenotypes of a *wrn*^{-/-} mutant zebrafish model or characteristic phenotypes of human chondrocytes deficient in WRN.

Response: We thanks the reviewer for bringing up this point. We agreed with the reviewer's suggestions and hereby included the following experiments to support our findings. We treated the *wrn* mutant zebrafish with G4 stabilizer followed previous reports (1). The optimal dose of G4 stabilizer that would not cause death or developmental malformations was firstly determined. One-cell stage *wrn*^{-/-} zebrafish embryos were injected with several concentrations of G4 stabilizer ranging from 0 ng/μl to 20 ng/μl. Embryos that were dead, deformed, and normal were evaluated at 24 hpf and 48 hpf. As results a and b below shown (summarized in Supplementary Fig. 13d-e), 0.05 ng/μl and 0.1 ng/μl were selected for subsequent experiments. The mRNA expression of *shox* was then evaluated at different timepoints (3 dpf, 7 dpf, and 14 dpf) and we noted that the mRNA expression of *shox* of *wrn*^{-/-} zebrafish decreased significantly compared to that of the wildtype zebrafish (Fig. c below, summarized in Supplementary Fig. 13f). Additionally, the expression of specific chondrocyte-associated markers (*sox9a*, *sox9b*, and *col2a1a*) was examined, and we showed that the mRNA expression of these genes was downregulated when treating with the G4 stabilizer in *wrn*^{-/-} zebrafish (Fig. d-g, summarized in Fig. 9n-p), indicating that the G4 structure inhibited *shox* expression and chondrogenesis. This finding was consistent with our results showing that destroying G4s could facilitate chondrogenesis (for data, please refer to Fig. 9k-m in the revised manuscript).

The related contents were also added in the revised manuscript: page 28 from line 696 to line 710 in the blue-colored font.

Reviewer 3---Question 2

G4 stabilizer treatment in *wrn* mutant zebrafish downregulated the expression of *shox* and inhibited chondrogenesis. **a-b.** Stacked bar graphs of zebrafish embryos in dead, deformed, and normal status with different G4 stabilizer concentration at 24 hpf (a) and 48 hpf (b). **c.** qRT-PCR measurement of *shox*. **d-g.** qRT-PCR measurement of *sox9a* (d), *sox9b* (f), and *col2a1a* (g) at different timepoints. Data are presented as the mean \pm S.D. Statistical analysis was performed using two-tailed unpaired Student's t-test. * $P < 0.05$, ** $P < 0.01$, *** $P < 0.001$. N=3 independent biological experiments.

Reference

(1). Moruno-Manchon, J. F. et al. Small-molecule G-quadruplex stabilizers reveal a novel pathway of autophagy regulation in neurons. *eLife*.**9**, e52283 (2020)

Methods for Reviewer 3-Questions2

G4 stabilizer treatment

G4 stabilizer was purchased from MedChemExpress (HY-15176A). 5nl of G4 stabilizer was injected with several concentrations ranging from 0 ng/ μ l to 20 ng/ μ l to determine the optimal dose. The embryos at 24 hpf and 48 hpf were collected for further

analysis.

Summary: The authors have addressed some points raised by my review, but it is still unclear to me if causal relationships among wrn status and G4-forming potential in shox, as it relates to bone phenotypes in the zebrafish or the human cellular phenotypes assayed, has been established.

Other changes:

In addition to addressing the comments by the reviewers, we have also made additional changes to improve the manuscript. These changes include:

1. We updated Fig. 1c with results using time points 3 dpf instead of 4 dpf because the timepoints we tested were mostly focused at 3 dpf, 7 dpf, and 14 dpf. We sought to keep the timepoints as consistent as possible throughout the entire manuscript.
2. We moved qRT-PCR and calcein staining results from previous supplementary Fig. 6 results to Fig. 4a, b, c, d, f. We hope that this change will facilitate a clearer presenting of the data showing the role of WRN helicase in chondrogenesis.
3. To provide strong experimental linkages between WRN and SHOX during chondrogenesis, we made new arranges of related data: the qRT-PCR and fluorescence *in situ* hybridization results were moved from previous supplementary Fig. 7f-g to main Fig. 5f-g; furthermore, we moved the heatmap of representative bone development, cell growth, and spinal cord development genes from previous main Fig. 5d-f to supplementary Fig. 6f-h.

We hope the additional data generated and re-organization of some of the figures could help improve the manuscript.

REVIEWERS' COMMENTS

Reviewer #3 (Remarks to the Author):

The authors have made reasonable responses to my remaining concerns. In addition they have strengthened the manuscript by adding more data with the new G4 ligand experiments. Overall, the manuscript is further improved.